# Molecular Phylogenetic and Comparative Genomic Analysis of *Pleurocordyceps fusiformispora* sp. nov. and *Perennicordyceps elaphomyceticola* in the Family Polycephalomycetaceae

**DOI:** 10.3390/jof10040297

**Published:** 2024-04-19

**Authors:** Zuoheng Liu, Yingling Lu, Dexiang Tang, Juye Zhu, Lijun Luo, Yue Chen, Hong Yu

**Affiliations:** 1Yunnan Herbal Laboratory, College of Ecology and Environmental Sciences, Yunnan University, Kunming 650504, China; lzh1313159@163.com (Z.L.); lyinglingua@163.com (Y.L.); tangdx1516@163.com (D.T.); zhujuye@mail.ynu.edu.cn (J.Z.); yunllj07@163.com (L.L.); cy106daytoy@163.com (Y.C.); 2The International Joint Research Center for Sustainable Utilization of Cordyceps Bioresources in China and Southeast Asia, Yunnan University, Kunming 650091, China

**Keywords:** new species, whole genome sequence, secondary metabolite, biosynthesis gene cluster, hyperparasitic fungi

## Abstract

Several *Pleurocordyceps* species have been reported as hyperparasitic fungi. A new species, *Pleurocordyceps fusiformispora*, and a known species, *Perennicordyceps elaphomyceticola*, are described here based on morphology and phylogenetic evidence from six genes (ITS, SSU, LSU, *TET1-α*, *RPB1*, and *RPB2*). *Pl. fusiformispora* differed from the other *Pleurocordyceps* species by producing flaky colonies, ovoid or elliptic α-conidia, and fusiform or long fusiform β-conidia. Both full genomes of *Pe. elaphomyceticola* and *Pl. fusiformispora* were sequenced, annotated, and compared. The antiSMASH and local BLAST analyses revealed significant differences in the number and types of putative secondary metabolite biosynthetic gene clusters, i.e., NPPS, PKS, and hybrid PKS–NRPS domains, between the two species. In addition, the putative BGCs of six compounds, namely ε-poly lysine, 4-epi-15-epi-brefeldin A, Monorden D/monocillin IV/monocillin VII/pochonin M/monocillin V/monocillin II, Tolypyridone, Piperazine, and Triticone DABFC, were excavated in the present study. This study motivates the use of heterologous expression and gene knockout methods to discover novel biologically active SMs from Polycephalomycetaceae.

## 1. Introduction

The classification system of *Cordyceps* sensu lato is widely accepted, with *Perennicordyceps* and *Pleurocordyceps* being sister genera within the family Polycephalomycetaceae [1]. Polycephalomycetaceae is a kind of fungus group with important application value and broad development and application prospects in medicine, agriculture, and other fields [1]. Xiao et al. [2] noted that *Pleurocordyceps* sp. exhibits significant potential for the production of a diverse range of secondary metabolites. Recent studies showed that *Pleurocordyceps nipponicus* has antibacterial [3,4] and antitumor activities [5]. Surapong et al. [6] isolated two compounds, Cordytropolone and Leptosphaerone A, from liquid cultures of *Pl. nipponicus*, and experimental studies showed that Cordytropolone has some antifungal activity.

Natural products, also known as secondary metabolites (SMs), have played an important role in the history of drug discovery and development [7]. As more and more microbial genomes are sequenced, bioinformatic analysis has revealed a vast resource of novel SMs. Genome mining is a new strategy of SM discovery based on gene cluster sequences and biosynthetic pathways. At the same time, it can directly associate the structures of SMs with synthetic pathways and facilitate the study of SM biosynthesis and combinatorial biosynthesis [8]. Some analysis tools and procedures (e.g., antibiotics and Secondary Metabolites Analysis Shell, antiSMASH; ClusterFinder; Antibiotic Resistant Target Seeker, ARTS) allow for the rapid and direct detection of biosynthetic gene clusters (BGCs), as well as the diversity of gene cluster families, such as polyketide synthases (PKSs), non-ribosomal peptide synthetases (NRPSs), and terpene synthases (TSs) [9]. More SM BGCs can be discovered through gene mining [10]. For instance, the sequence of amino acids of NpPKS3 in the lichenized fungus *Nephrmopsis pallescens* was 53% consistent with the type III PKS (CYSBs) of *Beauveria bassiana* [11]. Sayari et al. [12] found that all 20 genomes of Ceratocysti daceae showed extremely conserved PKS-III gene clusters containing homologous genes encoding the CHS. Wang et al. [9] discovered beauveriolide BGC in the *Cordyceps militaris* genome using bioinformatic analysis and then produced the compound via heterologous expression. Fumosorinone is a new 2-pyridone alkaloid isolated from *Cordyceps fumosorosea* [10]. The BGC of fumosorinone consisted of a hybrid PKS–NRPS, two cytochrome P450, a trans-enoyl reductase gene, and two other transcription regulatory genes [13]. In recent years, there has been little work on secondary metabolite gene clusters in *Perennicordyceps* and *Pleurocordyceps*.

In this study, the species *Pl. fusiformispora* Hong Yu bis and Z.H. Liu, D.X. Tang, Y.L. Lu, sp. nov. was first introduced. In order to discover more potential gene clusters of SMs, the whole genomes of *Pl. fusiformispora* and *Pe. elaphomyceticola* were sequenced and annotated and were used in gene mining studies. The potential of the Polycephalomycetaceae fungi to produce SMs was further analyzed.

## 2. Materials and Methods

### 2.1. Test Materials

*Pe. elaphomyceticola* was collected on 12 June 2022, suburb of Menghai County, Yunnan Province, China (21°58′10.09″ N, 100°27′30.89″ E, altitude: 1160.79 m). *Pl. fusiformispora* was collected on 23 July 2022, Wild Duck Lake Forest Park, Kunming City, Yunnan Province, China (25°6′24.98″ N, 102°50′12.28″ E, altitude: 2039.05 m). The voucher specimens were stored in Yunnan Herbal Herbarium (YHH) of Yunnan University, and the isolated strains were stored in Yunnan Fungal Culture Collection (YFCC) of Yunnan University.

### 2.2. Culture and Morphological Observations

Sexual morph observation was performed by photographing and measuring ascomata using an Olympus SZ61 stereomicroscope (Olympus Corporation, Tokyo, Japan). Freehand or frozen sections of the fruit body structure were placed in a solution of lactophenol cotton blue for microscopic study and photomicrography. The frozen sections were prepared using a freezing Microtome HM525NX (Thermo Fisher Scientific, Waltham, MA, USA). The micro-morphological characteristics of fungi (perithecia, asci, apical caps, and ascospores) were examined using Olympus CX40 and BX53 microscopes. The PDA solid media (20 g/L potato powder, 20 g/L glucose, 18 g/L agar powder, 1 L H_2_O (all chemicals and reagents were from Kunming City, China)) in which the colonies were cultivated were kept at room temperature (25 °C) for 40 days, and then the specimens were photographed and recorded using a Canon 750D camera (Canon Inc., Tokyo City, Japan) to observe the morphological properties of the colonies. The necessary microscope slide cultures were made according to [14], leaving them at 25 °C for 10 days. The colony was photographed and measured every fourth day. To characterize the strain, microscope slide cultures were generated by inoculating a small portion of the mycelium onto a 25 mm^2^ area of PDA medium (20 g/L potato powder, 20 g/L glucose, 18 g/L agar powder, 1 L H_2_O (all chemicals and reagents were from Kunming City, China)) block overlaid by a cover slip. Specimens were photographed and measured with an Olympus SZ61 stereomicroscope (Hamburg, Germany). Morphological observations and measurements were carried out using an Olympus CX40 microscope and a FEI QUANTA200 scanning electron microscope (Valley City, ND, USA). Fifty measurements were taken to collect the necessary information about the hypha, synnema, conidial mass, phialide, and conidium.

### 2.3. DNA Extraction, Polymerase Chain Reaction (PCR), and Sequencing

Specimens and axenic living cultures were prepared for DNA extraction. Total DNA was extracted using the CTAB method described by [15]. Polymerase chain reaction (PCR) was used to amplify genetic markers using the following primer pairs: LR0R/LR5 for small subunit nuclear ribosomal DNA (LSU) [16,17], EF1α-EF/EF1α-ER for translation elongation factor 1-α (*TEF1-α*) [18], RPB1-5′F/RPB1-5′R for partial RNA polymerase II largest subunit gene region (*RPB1*), RPB2-5′F/RPB2-5′R for partial RNA polymerase II second largest subunit gene region (*RPB2*) [19,20]. The PCR assay was completed with a final volume of 25 μL. Each reaction was composed of 2.5 μL of 10× Ex Taq Buffer (containing 2 mM MgCl_2_; TaKaRa, Kusatsu, Japan), 2 μL of 2.5 mM of each of the four dNTPs (TaKaRa), 1 μL of 10 μM of each primer, 0.25 μL of 5U Ex Taq DNA polymerase (TaKaRa), 2 μL of template DNA, and 17.25 μL ultrapure water. The LSU was amplified using the following PCR conditions: initial denaturation at 95 °C for 4 min and then 30 cycles of denaturation (each cycle at 95 °C for 1 min), annealing at 50 °C for 1 min, polymerization at 72 °C for 2 min, and a final extension at 72 °C for 8 min. PCR conditions for *TEF1-a*, *RPB1*, and *RPB2* were performed as previously described [21]. PCR products were separated by electrophoresis in 1.0% agarose gels, purified using the Gel Band Purification Kit (Bio Teke Co., Ltd., Beijing, China), and then sequenced on an automatic sequence analyzer (BGI Co., Ltd., Yantian, Shenzhen, China). When PCR products could not be sequenced directly, cloning was performed by the TaKaRa PMD™18-T vector system (TaKaRa Biotechnology Co., Ltd., Dalian, China).

### 2.4. Genome Sequencing and Assembly

*Pe. elaphomyceticola* and *Pl. fusiformispora* strains were cultured on PPDA solid medium (20 g/L potato powder, 10 g/L yeast powder, 20 g/L glucose, 18 g/L agar powder, 1 L H_2_O) at 25 °C for 60 days. The mycelium was transferred to PPA liquid medium (20 g/L potato powder, 10 g/L yeast powder, 20 g/L glucose, 1 L H_2_O) at 25 °C static cultivation for 2–3 months. Appropriate amounts of *Pe. elaphomyceticola* and *Pl. fusiformispora* mycelium were scraped, and total genomic DNA was extracted using the Plant DNA Isolation Kit (Foregene Co., Ltd., Chengdu, China), and then sequenced on an automatic sequence analyzer (BGI Co., Ltd., Wuhan, China) using the same primers as used in amplification. Sequencing data contain some low-quality reads with joints, which can cause significant interference in subsequent information analysis. Illumina NovaSeq 2000 (Nanopore, Wuhan, China) high-throughput sequencing platform, used for sequencing a gene library with 400 bp insertion fragment, was used with a sequencing mode of paired-end and 2 × 150 bp. The fastp ([https://github.com/OpenGene/fastp], accessed on 14 July 2023) was used to filter the raw reads, discard low-quality reads, and obtain clean data. In order to ensure the quality of subsequent information analysis, it was necessary to further filter the raw data to generate high-quality sequences. The standards for data filtering mainly include the following points: joint contamination removal, using AdapterRemoval (v2.0) [22] to remove joint contamination from the 3 ‘end, and quality correction, using SOAPec (v2.0) software to perform quality correction on all reads based on Kmer frequency, with a Kmer setting of 17 used for correction. A5-MiSeq and SPAdes were used to construct contig and scaffold. Finally, the assembly effects of contigs and scaffolds were evaluated using pilon v1.18 [23] software.

### 2.5. Gene Prediction and Annotation

A combination of de novo gene prediction, transcript mapping, and homologous searches were used for gene prediction. Based on the existing database on gene function and metabolic pathways, the predicted genes were annotated by BLAST search, including Kyoto Encyclopedia of Genes and Genomes (KEGG), NCBI non-redundant protein sequences (NR), Gene Ontology (GO), Cluster of Orthologous Groups of eukaryotic complete genomes (KOG), Pfam, and Interpro.

### 2.6. Analysis of Secondary Metabolite Biosynthesis Gene Cluster

The antiSMASH (https://antismash.secondarymetabolites.org/, accessed on 1 September 2023) online program was used to perform gene cluster prediction at the level of genomic scaffolds of *Pe. elaphomyceticola* and *Pl. fusiformispora* complex. Based on antiSMASH-detected scaffolds with gene clusters, the online program FGENESH (www.softberry.com/, accessed on 26 September 2023) was used to predict gene structures using *Ophiocordyceps sinensis* as a parameter. To obtain the domain, the PKS/NRPS online program (https://nrps.igs.umaryland.edu/, accessed on 11 October 2023) was used to predict gene clusters in contigs where NRPS/PKS genes were located. At the same time, the online program Protein BLAST (https://blast.ncbi.nlm.nih.gov/, accessed on 25 October 2023) was used for NRPS/PKS genes of contig protein ratio analysis.

### 2.7. Cluster Analysis

The polygene nucleotide sequences (ITS, SSU, LSU, *TEF-1a*, *RPB1*, *RPB2*) were downloaded from NCBI (https://www.ncbi.nlm.nih.gov/, accessed on 25 August 2023) and compared with these sequences using the Clustal W program in the MEGA5.0 software for multi-sequence comparison [24,25]. Based on a six-gene dataset, the software PhyloSuite (v1.2.2 Win) was used to construct phylogenetic tree of maximum likelihood (ML) and Bayesian inference (BI). In addition, we used the method from the online program IQ-TREE web server (http://iqtree.cibiv.univie.ac.at/, accessed on 15 July 2023) since the report 1000 and the rest of the default parameter to construct an ML cluster analysis tree of NRPS or hybrid PKS–NRPS proteins of *Pe. elaphomyceticola*, *Pl. fusiformispora*, and other fungi.

## 3. Results

### 3.1. Phylogenetic Analysis

Based on the joint matrix of nucleotide sequences of ITS, SSU, LSU, *TEF1-α*, *RPB1*, and *RPB2*, the molecular phylogenetic tree of Polycephalomycetaceae was reconstructed by ML and BI. The total length of the concatenated dataset of six genes across the 86 samples was 5303 bp, including 554 bp for ITS, 920 bp for SSU, 911 bp for LSU, 975 bp for *TEF-1α*, 739 bp for *RPB1*, and 1,204 bp for *RPB2*. The phylogenetic tree, represented by *Tolypocladium ophioglossoides* NBRC 106330 and *T. ophioglossoides* NBRC 100998 as the outgroup taxa, consisted of three clades, i.e., *Polycephalomyces* clade, *Perennicordyceps* clade, and *Pleurocordyceps* clade, with a total of 86 sequences (Table 1), of which 4 were self-detected. The matrix had 2646 distinct patterns, 1669 parsimony-informative sites, 666 singleton sites, and 13,982 constant sites. ModelFinder was used to select the best-fitting likelihood model (GTR+F+I+G4) for ML analyses and BI analyses according to the Akaike information criterion (AIC). The ML and BI tree with the best score found was −40,864.213, and the total tree length was 1.098. The generic-level relationships of ML and BI trees were topologically similar. The tree was visualized with its maximum likelihood bootstrap proportions (ML-BS) and Bayesian posterior probability (BI-BPP) in Figtree v.1.4.3 (Figure 1).

From the perspective of a six-gene phylogenetic tree, the three branches of the Polycephalomycetaceae receive high support rates. The cluster of *Pl. fusiformispora* collected and described in this study was in the adjacent branches of *Pl. sinensis*. *Pl. fusiformispora* and *Pl. sinensis* were sister species to each other and formed a separate clade (BS = 96%, BPP = 0.89). *Pe. elaphomyceticola* and *Pe. prolifica* were sister species to each other and formed a separate clade (BS = 99%, BPP = 1). This was consistent with the results [1].

### 3.2. Taxonomy

#### 3.2.1. *Pleurocordyceps fusiformispora* Hong Yu bis and Z.H. Liu, D.X. Tang, Y.L. Lu, sp. nov. Figure 2

##### MycoBank: MB 851478

Etymology: The species name refers to the production of fusiform or long fusiform conidia during the asexual phase. Hyperparasitic on *Ophiocordyceps* sp. (Ophiocordycipitaceae) and on insects buried in soil. Sexual morph: undetermined. Asexual morph: Synnemata measure 1.2–1.5 cm long by 0.1–0.5 mm wide, clavate, capitate, crowd on the insect body and *Ophiocordyceps* sp., unbranched, white to yellowish, with or without fertile head at the apex. Stipes 2.1–2.8 cm long, 0.7–1.0 mm wide. Colonies on PDA grow slowly, attaining a diameter of 5.1–5.5 cm in 40 days at 25 °C, clustered, white–yellow, and reverse dry yellow. Synnemata emerge after 20 days, flaky, branched, 1.4–1.9 cm long and 0.3–0.5 cm wide, showing radiating distributions. Phialides exist in two types: α- and β-phialides. Both types of phialide often produce new phialides at their own apices or yield catenulate β-conidia, collarettes not flared, periclinal thickening not visible. The α-phialides are acropleurogenous on conidiophores and solitary on hyphae, narrow lageniform or subulate, taper abruptly from the base to the apex, 8.42–20.9 μm long, 1.3–2.9 μm wide at the base, and 0.6–1.6 μm wide at the apex. The β-phialides are solitary on hyphae, lanceolate, taper gradually from the base to the apex, 8.7–14.8 μm long, 2.5–3.1 μm wide at the base, and 0.9–1.1 μm wide at the apex. α-conidia are ovoid or elliptic and occur in the conidial mass on the agar or on the final portion of synnema, 2.9–4.8 × 1.3–3.1 μm. β-conidia are fusiform or long fusiform and are produced on the surface mycelium of colony, usually in chains on a phialide, 2.8–4.4 × 1.7–2.8 μm.

Material examined: Wild Duck Lake Forest Park, Kunming City, Yunnan Province, China, hyperparasitic on *Ophiocordyceps* sp. (Ophiocordycipitaceae) and on insects buried in soil. Collected on 23 July 2022, holotype: YFCC 07239279, paratype: YFCC 07239280, other collections: YFCC 07319281.

Notes: *Pleurocordyceps fusiformispora* was sister to the clade formed by *Pl. sinensis* (Figure 1: 94% ML/0.88 PP). *Pl. fusiformispora* differs from the other *Pleurocordyceps* species by producing flaky colonies, their α-conidia are ovoid or elliptic, and their β-conidia are fusiform or long fusiform (Appendix A).

**Figure 2 jof-10-00297-f002:**
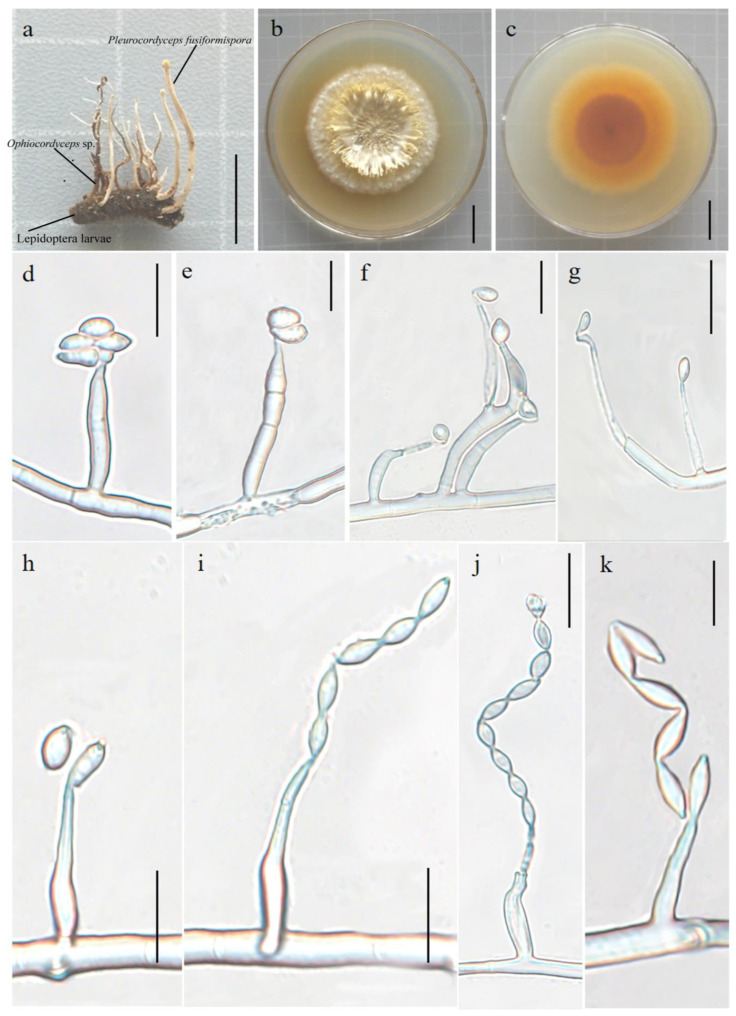
*Pleurocordyceps fusiformispora* (holotype: YFCC 07239279). (**a**) Overview of *Pleurocordyceps fusiformispora* and its host. (**b**,**c**) Colony, obverse and reverse. (**d**–**h**) α-phialides and α-conidia. (**i**–**k**) β-phialides and β-conidia. Scale bars: (**a**) = 1cm; (**b**,**c**) = 2 cm; (**d**–**f**,**h**–**j**) = 10 µm; (**g**) = 20 µm; (**k**) = 5 µm.

#### 3.2.2. *Perennicordyceps elaphomyceticola* (WY Chuang, Ariyawansa, Jiuein Yang and Stadler) Y.P. Xiao and K.D. Hyde, comb. nov [1], Figure 3 and Figure 4

Synonymy: *Polycephalomyces elaphomyceticola* W.Y. Chuang, H.A. Ariyaw., J.I. Yang and Stadler, in Yang, Stadler, Chuang and Ariyawansa, *Mycol. Progr*. **2020,** 19(1): 102.

Etymology: The specific epithet *elaphomyceticola* is based on the host genus from which the fungus was isolated. Parasitic on *Elaphomyces* sp. (Elaphomycetaceae) from soil. Sexual morph: stromata 5.1–6.2 cm long, 0.5–0.7 cm wide, cylindrical, solitary or several, branched, the color gradually becomes lighter towards the apex, yellow to dark yellow to light yellow, hard. Fertile heads: 1.5–2 cm long, 0.1–0.3 cm wide, branched, dark yellow to light yellow, upper surface roughened. Perithecia: 259–519 × 152–291 μm, superficial, ovoid to ellipsoid. Asci: 164–173 × 3.1–5.5 μm, hyaline, cylindrical. Apical cap: 2.1–3.5 × 3.6–4.2 μm, thin, hyaline. Ascospores: 55.1–105 × 0.8–1.2 μm, irregular multiseptate. Secondary spores: 0.8–1.1 × 0.6–0.8 μm globose to cylindrical, one-celled, hyaline, smooth-walled. Asexual morph: (see Figure 4) colonies on PDA 3.7–4.0 cm in diameter after 40 days at 25 °C, usually verrucose, white to orange-yellow. On the reverse appear vague concentric rings, black–brown in the center and maple-colored at the edge. Phialides develop from the edge of the colony and conidial mass of the synnema. Phialides: cylindrical to subulate at the base, occur directly on the aerial hyphae, 16.8–31.9 µm in length, taper gradually from 2.0–3.8 µm at the base to 0.4–1.1 µm at the apex, generating a single or lumpy conidia. Conidia: oval, 3.2–5.1 × 0.4–1.2 µm.

Material examined: Suburb of Menghai County, Yunnan Province, China. Parasitic on *Elaphomyces* sp. in soil, 12 June 2022, YFCC 06129282, Hong Yu.

**Figure 3 jof-10-00297-f003:**
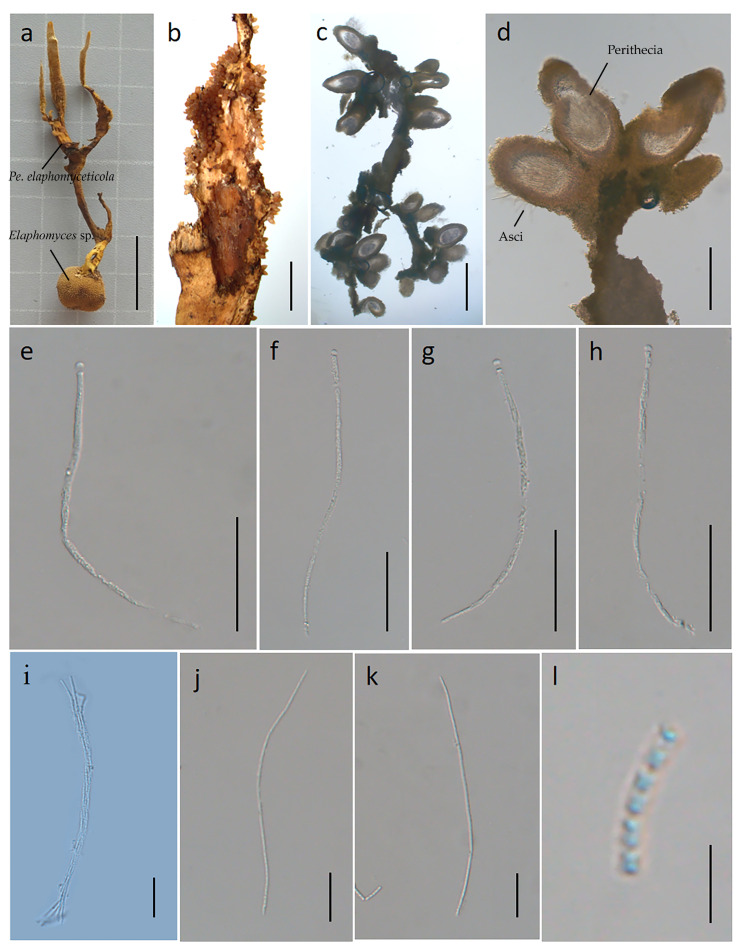
Sexual morph of *Perennicordyceps elaphomyceticola* (YFCC 06129282). (**a**) Stromata emerging from infected *Elaphomyces* sp. (**b**) Fertile head of ascostroma. (**c**) Vertical section of stroma. (**d**) Perithecia. (**e**–**h**) Asci. (**i**) Asci and ascospore. (**j**,**k**) Ascospore. (**l**) Secondary ascospores. Scale bars: (**a**) = 2 cm; (**b**) = 5000 µm; (**c**) = 500 µm; (**d**) = 200 µm; (**e**–**h**) = 50 µm; (**i**–**k**) = 20 µm; (**l**) = 5 µm.

**Figure 4 jof-10-00297-f004:**
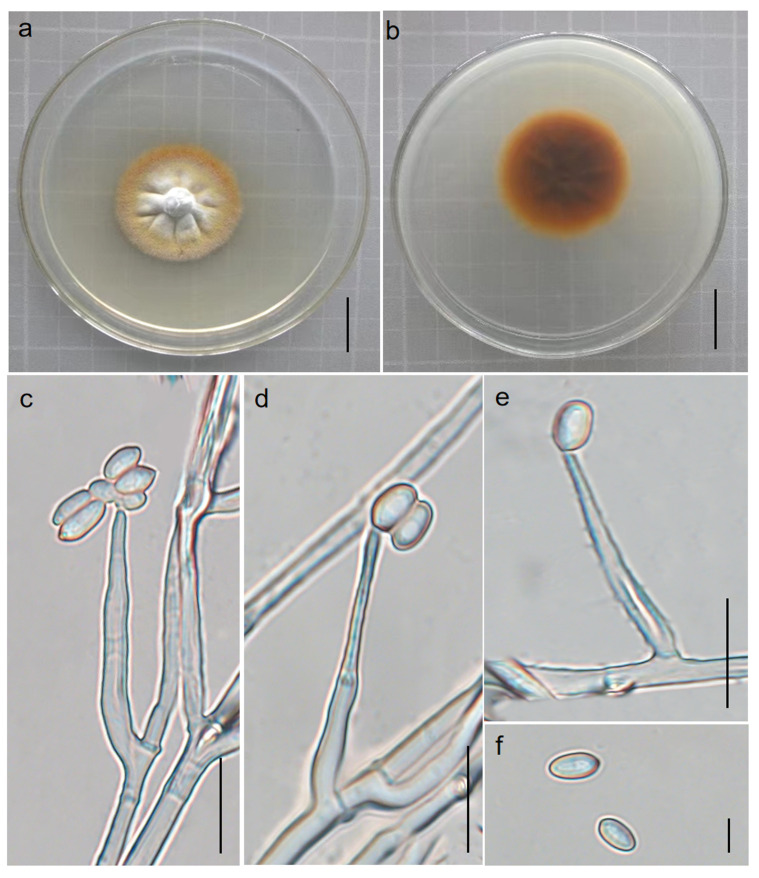
Asexual morph of *Perennicordyceps elaphomyceticola* (YFCC 06129282). (**a**,**b**) Colony obverse and reverse. (**c**–**e**) Phialide and conidia. (**f**) Conidia. Scale bars: (**a**,**b**) = 2 cm; (**c**–**e**) = 10 µm; (**f**) = 5 µm.

### 3.3. Basic Genomic Characteristics of Perennicordyceps elaphomyceticola and Pleurocordyceps fusiformispora

#### 3.3.1. Genome Sequencing and Assembly

The Illumina sequencing produced 24,006,990 raw reads and 3,600,527,330 high-quality reads of *Pl. fusiformispora* (Appendix A). The size of the genome *Pe. elaphomyceticola* was 31.51 Mb, containing 48.79% GC content and 7794 protein-coding genes along with 103 tRNA, 18 rRNA, and 52 ncRNA. Among the genome size of *Pl. fusiformispora* was 33.02 Mb, containing GC content, protein-coding genes, tRNA, rRNA, and ncRNA, which were 50.08%, 8804, 100, 23 and 3, respectively. The results showed that the genome assembly of these two species was of good quality. In phylogeny, *Pleurocordyceps* and *Perennicordyceps* were sister genera (Figure 1). According to the above data, these results suggested that the closer the relationship between genera, the probable smaller the differences in genome size, GC content, and total gene number.

#### 3.3.2. Genome Annotation

The *Pe. elaphomyceticola* species were identified using the KEGG databases (3509 genes/45.02%), the EggNOG databases (7196 genes/92.32%), the NR databases (7536 genes/96.68%), the GO database (5302 genes/68.02%), and the Pfam databases (5927 genes/76.04%). However, an analysis of 6706 non-redundant *Pl. fusiformispora* genes in a publicly available protein sequence database produced mixed results; these were from the KEGG databases (3723 genes/42.28%), the EggNOG databases (8183 genes/92.94%), the NR databases (8566 genes/97.29%), the GO database (5112 genes/69.50%), and the Pfam databases (6726 genes/76.39%) (Appendix A). The KEGG functional classification of *Pe. elaphomyceticola* and *Pl. fusiformispora* indicated protein families for genetic information processing, signaling and cellular processing, and signal transduction (respectively, Figure 5b,e). In addition, the rich genetic information and diversity of signaling proteins may facilitate more efficient information exchange and secondary metabolism.

According to the EggNOG database, *Pl. fusiformispora* and *Pe. elaphomyceticola* captured more predicted genes as “Function unknown”, “Posttranslational modification, protein turnover, chaperones”, and “Carbohydrate transport and metabolism” (Figure 5a,d). However, a wide variety of carbohydrate metabolism and post-translational events suggested that they enhanced the bioactivity and energy conversion efficiency of regulatory proteins. GO annotation of the genomes of both species showed that translation, protein transport, and glucose metabolism were abundant genes in biological processes (Figure 5c,f). Furthermore, the GO annotation revealed integral components of the membrane, nucleus, and cytoplasm from the cellular component and ATP binding, metal ion binding, and zinc ion binding from the molecular function. The two species of *Pe. elaphomyceticola* and *Pl. fusiformispora* were wild strain, in which many metabolic genes might be involved in signal transduction.

#### 3.3.3. Additional Annotation

##### Carbohydrate-Active Enzymes (CAZy)

Enzymes that played an important role in carbohydrate modification, biosynthesis, and the degradation of fungi were carbohydrate-active enzymes (CAZy) [36], which are present in a database of carbohydrate-active enzymes and a special database of carbohydrate enzymes [37]. It was shown that *Pe. elaphomyceticola* (Figure 6a) and *Pl. fusiformispora* (Figure 6b) had many glycoside hydrolases (GHs), glycosyl transferases (GTs), and auxiliary activities (AAs), leading us to speculate that *Pe. elaphomyceticola* and *Pl. fusiformispora* possibly breakdown complex carbohydrates and capture more energy.

##### Pathogen–Host Interactions (PHI)

The full name of PHI is Pathogen Host Interactions Database, which is a database of pathogen–host interactions, mainly derived from fungi, oomycetes, and bacterial pathogens. The infected hosts include animals, plants, fungi, and insects [38]. This database plays an important role in searching for target genes for drug intervention, and it also includes antifungal compounds and corresponding target genes. The results showed that *Pe. elaphomyceticola* (Figure 7a) and *Pl. fusiformispora* (Figure 7b) had reduced virulence and the loss of pathogenicity genes. The primary annotated genes of *Pe. elaphomyceticola* and *Pl. fusiformispora* from the PHI database were used to reduce virulence and do not affect pathogenicity, indicating that *Pe. elaphomyceticola* and *Pl. fusiformispora* are not highly pathogenic strains.

### 3.4. Analysis of Secondary Metabolite Biosynthesis Gene Cluster

#### Overview of Twelve Genomic BGCs of *Pe. elaphomyceticola* and *Pl. fusiformispora*

AntiSMASH and local BLAST analyses revealed that *Pe. elaphomyceticola* (39) and *Pl. fusiformispora* (50) had different putative SM BGCs (Appendix A). *Pl. fusiformispora* and *Pe. elaphomyceticola* predicted that the largest number of SM BGC types were NRPS, with 16 and 15, respectively. *Pl. fusiformispora* had five hybrids, PKS + NRPS, and *Pe. elaphomyceticola* had two. Both *Pl. fusiformispora* and *Pe. elaphomyceticola* had five terpenes and two hybrids, NRPS + other genes. *Pl. fusiformispora* and *Pe. elaphomyceticola* had seven and six other genes, respectively. For the predicted PKSs, *Pl. fusiformispora* had fifteen PKSs, while *Pe. elaphomyceticola* had nine. The *Pl. fusiformispora* genome had fifteen PKSs, including ten HR–PKSs, three non-reducing (NR) PKSs, and two partially reducing (PR) PKSs. The genome of *Pe. elaphomyceticola* had nine PKSs, including four NR-PKSs, four HR-PKSs, and one PR-NRPS. These results indicated that the number and type of SM BGCs obtained in different species vary.

The predicted BGCs had different levels of genetic homology compared by known cluster in the MIBiG database, with *Pl. fusiformispora* having the highest homology (42%), followed by *Pe. elaphomyceticola* (41.02%). Both *Pe. elaphomyceticola* and *Pl. fusiformispora* were predicted to catalyze the synthesis of ε-poly-lysine and Squalestatin S1 gene clusters. *Pl. fusiformispora* was predicted to catalyze the synthesis of Fusaric acid, Choline, Cyclopiazonic acid, IlicicolinH, Lucilactaene, Ascochlorin, YWA1, Tolypyridone, Monorden D/monocillin IV/monocillin VII/pochonin M/monocillin V/monocillin II, Fusarin C, AKML A-C, UNLL-YC2Q1O94PT, 4-epi-15-epi-brefeldin A, and Ergotamine (Appendix A). In contrast, the BGCs *Pe. elaphomyceticola* predicted biosynthesis genes that catalyzed the synthesis of piperazine compound **1** and piperazine compound **2**, Triticone DABFC, and Viriditixin. Further, the types and amounts of compounds predicted by catalytic synthesis differ between the two species.

Several BGCs of *Pe. elaphomyceticola* and *Pl. fusiformispora* were 100% similar to MIBiG sequences. Purev et al. [39] found that ε-poly-lysine had antifungal activity. The predicted *Pe. elaphomyceticola* Region 13.1 and *Pl. fusiformispora* Region 21.4 were responsible for ε-poly-lysine (Figure 8). The local BLAST comparison demonstrated that ε-poly-lysinehas requires enzymes with the A-P-T structural domain, PKc-like super family, GAT_1 super family, and MFS. Both species have the potential to synthesize ε-poly-lysinehas. Zhang et al. [40] studied genome mining of in-house strains using hybrid PKS–NRPS as a query and identified an endophyte *Tolypocladium* sp. 49Y, which possesses a potential 4-hydroxy pyridone biosynthetic gene cluster. Moreover, heterologous expression in *Aspergillus oryzae* NSAR1 revealed that this gene cluster was functional and able to produce a rare type of 4-hydroxy pyridones called Tolypyridone. The predicted Region 19.2 and Region 19.3 of *Pl. fusiformispora* might be responsible for Tolypyridone biosynthesis (Figure 9). Qin et al. [41] proved that Monorden D/monocillin IV/monocillin VII/pochonin M/monocillin V/monocillin II compounds have modest antibacterial activities. By comparison, *Pl. fusiformispora* Region 21.2 and Region 21.3 had highly homologous regions, being highly similar to the synthetic Monorden D/monocillin IV/monocillin VII/pochonin M/monocillin V/monocillin II gene cluster (Figure 10). These results indicated that the catalytic synthesis of NR-PKS (SAT-KS-AT-PT-ACP-Te), HR-PKS (KS-AT-DH-ER-KR-ACP), FixC super family, MFS_Tpol_MDR_like, and CYP60B-like were necessary genes. The results showed that (Figure 11) *Pe. elaphomyceticola* Region 61.1 was like the gene cluster DS231617.1 that catalyzed the synthesis of Triticone ABFC. However, *Pl. fusiformispora* did not have the cluster of Triticone ABFC. *Pe. elaphomyceticola* Region 9.1 was highly similar to the known gene cluster that catalyzed the synthesis of piperazine (Figure 12). The slight difference was that one gene in *Pe. elaphomyceticola* Region 9.1 had dehydrogenase and CYP503A1-like functions. Moreover, KJ728786.1 synthesized 4-epi-15-epi-brefeldin A (Figure 13). The core gene structural domain (KS-AT-ER-KR-ACP) in *Pl. fusiformispora* Region 68.1 was similar to KJ728786.1. However, *Pl. fusiformispora* lost some of its modified genes during evolution. It was speculated that the position direction of key enzymes and modified genes was different when different genera were in the compound. At the same time, there will be gene fusion and gene loss in the process of evolution.

### 3.5. Cluster Analysis

The protein clustering results of NRPS and hybrid PKS–NRPS from *Pe. elaphomyceticola* and *Pl. fusiformispora* were compared to other fungal NRPS and hybrid PKS–NRPS. It was showed that *Pe. elaphomyceticola* Region 13.1 and *Pl. fusiformispora* Region 21.4 aggregated with *Epichloe festucae* (BBU42014.1), with being catalyzed the biosynthesis of ε-poly-lysine or its analogs (Figure 14). Moreover, *Pe. elaphomyceticola* Region 9.1 might catalyze the biosynthesis of piperazine compound **1** and piperazine compound **2** or its analogs. *Pl. fusiformispora* Region 19.2–19.3 and *Tolypocladium* sp. 49Y (QPC57090.1) were clustered on a branch that produced Tolypyridone. It was speculated that *Pl. fusiformispora* Region 19.2–19.3 presumably catalyzed the synthesis of Tolypyridone or its analogs. *Pl. fusiformispora* Region 21.2 and Region 21.3 clustered with *Chlamydosporia* 170 (OAQ63055.1) and (OAQ63050.2) catalyzed Monorden D/monocillin IV/monocillin VII/pochonin M/monocillin V/monocillin II and probably produced Monorden D/monocillin IV/monocillin VII/pochonin M/monocillin V/monocillin II or its analogs. The *Pl. fusiformispora* Region 68.1 converged with the *Penicillium brefeldianum* (KJ728786.1) to form an independent branch, which might synthesize 4-epi-15-epi-brefeldin A or its analogs. *Pe. elaphomyceticola* Region 61.1 clustered with the hybrid PKS-NRPS protein and possibly catalyzed Triticone DABFC synthesis in *Pyrenophora tritici-repentis* Pt-1C-BFP (DS231617.1) (Figure 15). Moreover, *Pe. elaphomyceticola* Region 61.1 possibly catalyzed Triticone DABFC or its analogs.

### 3.6. Synteny Analysis

The scaffolds containing the SM BGC in the genomes of *Pe. elaphomyceticola* (39) and *Pl. fusiformispora* (50) were subjected to synteny analysis. The scaffolds where the SM BGC are located are divided into more than 10 collinear blocks, and there may be rearrangement (Figure 16). From bottom to top, they are *Pl. fusiformispora* and *Pe. elaphomyceticola*, respectively.

## 4. Discussion

Herein, we describe a new species of *Pl. fusiformispora* using a combination of morphology and phylogeny. The newly established species distinctly form independent clades in the phylogenetic tree (Figure 1). Morphologically, the new species *Pl. fusiformispora* is similar to *Pl. aurantiacus*, *Pl. agarica*, *Pl. heilongtanensis*, *Pl.lanceolatus*, *Pl. marginaliradians*, *Pl. nutansis*, *Pl. vitellina*, and *Pl. yunnanensis* in that they have two types of phialides and conidia. However, the species *Pl. fusiformispora* has colonies that are flaky, α-conidia that are ovoid or elliptic, and β-conidia that are fusiform or long fusiform and differ from those of other species of *Pleurocordyceps* [1,24,25,42]. The discovery of the new species of *Pl. fusiformispora* adds to the diversity of the genus *Pleurocordyceps*.

Xiao et al. [1] noted five new samples (*Pl. sinensis* GACP 19-2301, *Pl. sinensis* MFLU 21-0269, *Pl. sinensis* MFLU 21-0268, *Pl. sinensis* GACP 20-0865, and *Pl. sinensis* GACP 20-2304), each of which is parasitic on a different host clustered with *Pl. sinensis*. These hosts were clavicipitoid fungi and constituted new hosts for *Pl. sinensis*. In addition, Xiao et al. [1] introduced *Pl. nutansis* as a new species under *Pleurocordyceps* and identified the host as *Ophiocordyceps nutans*. However, in the phylogenetic tree, there was a disorder in the systematic position of these two different hosts, which might be due to genetic differences caused by different strains.

Some species of Polycephalomycetaceae have also been reported as hyperparasitic fungi; these species were *Cordyceps*, *Elaphomyces*, *Hirsutella*, *Myxomycetes*, and *Ophiocordyceps* [24,25,32,43,44]. Through this investigation, we have observed that *Pl. fusiformispora* can parasitize both Lepidoptera larvae and *Ophiocordyceps* sp. We speculate that *Pl. fusiformispora* may also exhibit a hyperparasitism phenomenon. In addition, Xiao et al. [1] identified that *Pe. elaphomyceticola* can parasitize *Ophiocordyceps* sp., suggesting that *Pe. elaphomyceticola* probably also exhibits a hyperparasitism phenomenon. Most species in the genus *Pleurocordyceps* and *Perennicordyceps* exhibit a hyperparasitism phenomenon [1]. At present, little is known about the hyperparasitic mechanisms of the *Pleurocordyceps* and *Perennicordyceps* groups. Further research is needed on the relationship between parasitic species and their hosts.

In this paper, we present the basic genomic characteristics of *Pe. elaphomyceticola* and *Pl. fusiformispora*. The results showed that there was a certain difference in genome size, GC content, N50, and total number of genes. The BGCs of SMs that might be associated with hyperparasitism were analyzed through a gene mining analysis of *Pe. elaphomyceticola* and *Pl. fusiformispora* complexes. The estimated number of SM BGCs in the genomes of *Pe. elaphomyceticola* and *Pl. fusiformispora* were 39 and 50, respectively. However, we found that *Pl. fusiformispora* has a large number of PKS and HR-PKS. The number of NRPS, other and PKS, and other in each species was small or even absent. It was also unknown which functions the existing *Pe. elaphomyceticola* and *Pl. fusiformispora* might perform in their life activities.

Through genome mining, *Cordyceps militaris* was found likely to have existing NRPS and PKS [45]. In this study, many putative NRPS, PKS, and hybrid NRPS–PKS have been obtained. Many of the compounds were more similar to known gene clusters, such as Ochrindole A, Tubulysin A, Viriditixin, Cryptosporioptide BAC, Pyripyropene A, Leucinostatin A/Leucinostatin B, Emericellamide AB, Squalestatin S1, Sirodesmin PL, Pyripyropene, Ansaseomycin AB, Fusaric acid, Choline, Cyclopiazonic acid, Ilicicolin H, Lucilactaene, Ascochlorin, YWA1, Fusarin C, Betaenone C probetaenone I stemphyloxin II, AKML BD AC, Patulin, UNLL-YC2Q1O94PT, Scytophycin, Mangicol A, Ergotamine, and Oxaleimide C, could not be analyzed further because there were no modified genes, known gene clusters, key enzymes, indoles, or terpenes. The unique NR-PKS domains of *Pl. fusiformispora* include SAT-KS-AT-PT-ACP-Te, KS-AT-DH-MT, and PT-ACP-Te. At the same time, *Pe. elaphomyceticola* also had the unique NR-PKS domain of KS-AT-DH. They also had a common domain, NR-PKS (PKS-AT). The unique and common domains play an important role in the life cycle of each species. To exert biological functions, corresponding compounds need to be catalyzed, reflecting genetic differences between species.

Purev et al. [39] isolated the fungal gene “epls” encoding ε-poly lysine synthetase and confirmed that overexpression of epls in the different strain *Epichloë festucae* Fl1 resulted in the production of shorter ε-PL with 8–20 lysine, which exhibited a comparable antifungal activity to the longer one. In this study, the whole genome sequences of two species (*Pe. elaphomyceticola* Region 13.1 and *Pl. fusiformispora* Region 21.4) were present in homologous regions and were as high as 100% similar to the ε-PL gene sequence produced in MIBiG database. Therefore, the two species of *Pe. elaphomyceticola* and *Pl. fusiformispora* might produce ε-PL synthetase compounds, and their antifungal activities affect the growth of fungal hosts.

Zhang et al. [40] enacted genome mining of in-house strains using polyketide synthase-nonribosomal peptide synthase as a query and identified an endophyte, *Tolypocladium* sp. 49Y, which possessed a potential Tolypyridone biosynthetic gene cluster. Heterologous expression in *Aspergillus oryzae* NSAR1 revealed that this gene cluster is functional and able to produce a rare type of 4-hydroxy pyridones called Tolypyridone. And it was also found that Tolypyridone had antifungal activity. The whole genome sequences of *Pl. fusiformispora* were present in homologous Regions and were as high as 100% similar to the Tolypyridone gene sequence produced in the MIBiG database. *Pl. fusiformispora* may produce Tolypyridones compounds during infection with a parasite (*Ophiocordyceps* sp.), thereby inhibiting the normal development of *Ophiocordyceps* sp.

Qin et al. [41] discovered that the compounds monocillin VI/monocillin VII/monocillin II/monorden D/monocillin IV/monocillin V/pochonin M had moderate antibacterial activity. It had also been proven that these compounds had the potential to control bacterial diseases. The whole genome sequences of *Pl. fusiformispora* were present in homologous regions and were as high as 100% similar to the monocillin VI/monocillin VII/monocillin II/monorden D/monocillin IV/monocillin V/pochonin M gene sequence produced in MIBiG database. As a result, *Pl. fusiformispora* might produce monocillin VI/monocillin VII/monocillin II/monorden D/monocillin IV/monocillin V/pochonin M. Further investigation of the compounds presented in this study is needed to understand the functionality of *Pl. fusiformispora*.

The triticones, also known as spirostaphylotrichins, were characterized by a spirocyclic γ-lactam core structure. The triticones A to F were first purified from Ptr culture filtrates but were then found, along with an additional 18 other triticone compounds, in five other ascomycete fungi, *Staphylotrichum coccosporum*, *Curvularia pallescens*, *Pyrenophora seminiperda*, *Cochliobolus lunatus*, and *Bipolaris* spp. [46,47,48,49,50,51,52]. Triticone A and B are known to be phytotoxic, producing yellowish-brown lesions following leaf puncture assays on a range of hosts, including wheat. Triticones C and D were described as weakly active, and triticones E and F as inactive [49]. Rawlinson et al. [53] found that PKS–NRPS (KS-AT-DH-KR-C-A-T-R) was the key enzyme catalyzing triticones. AntiSMASH and local BLAST analyses showed that *Pe. elaphomyceticola* Region 61.1 and similar homologous regions also exist. The whole genome sequences of *Pe. elaphomyceticola* were present in homologous Regions and were as high as Triticone DABFC 57% similar to the Triticone DABFC gene sequence produced in MIBiG database. Consequently, *Pe. elaphomyceticola* might produce Triticone DABFC compounds. During the process of infecting the host, the *Pe. elaphomyceticola* species might synthesize Triticone DABFC compounds, which inhibit the growth of the host.

Forseth et al. [54] discovered the structural formula of the piperazine compound **1** and piperazine compound **2**. The whole genome sequences of *Pe. elaphomyceticola* were present in homologous regions as high as 50% similar to the piperazine compound **1**, piperazine compound **2** gene sequence produced in MIBiG database. Consequently, *Pe. elaphomyceticola* might produce piperazine compound **1** and piperazine compound **2**. The presence of this compound in *Pe. elaphomyceticola* requires further study of its functionality.

Brefeldin A is a unique fungal metabolite of a 13-membered macrocyclic lactone ring [55] and shows a wide range of interesting biological activities, including an inhibitory effect on virus multiplication [56,57]. Zabala et al. [58] found that HR-PKSs (KS-AT-DH-ER-KR-ACP) were the key enzymes catalyzing Brefeldin A (BFA). Similar homologous Regions 68.1 of *Pl. fusiformispora* were also present in both the antiSMASH and local BLAST analyses. Only the CYP-modified gene was present, and the other modified genes were lost. Consequently, *Pl. fusiformispora* might produce Brefeldin A (BFA) compounds. The presence of this compound in *Pl. fusiformispora* requires further investigation of its functionality.

In this study, six compounds—ε-poly lysine, 4-epi-15-epi-brefeldin A, Monorden D/monocillin IV/monocillin VII/pochonin M/monocillin V/monocillin II, Tolypyridone, Piperazine, and Triticone DABFC—were discovered from the two species of *Pe. elaphomyceticola* and *Pl. fusiformispora*. In these species, the discovery of novel compound BGCs could lead to the development of new applicable antifungals. In addition, the clusters of genes that different species catalyzed to synthesize the same compound were different. It was conjectured that there might be some degree of horizontal gene transfer among these species, that the direction and location of these gene sequences might be variable, and that gene mosaicism, gene loss, or addition might occur among different species.

## Figures and Tables

**Figure 1 jof-10-00297-f001:**
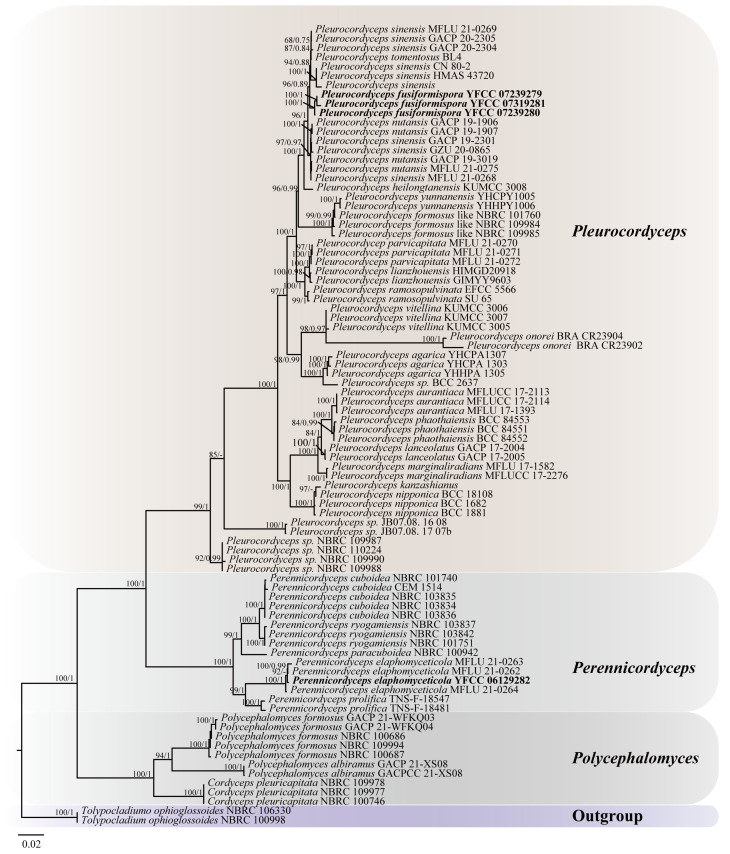
Phylogenetic tree of Polycephalomycetaceae was constructed, based on the concatenation of ITS, SSU, LSU, *TEF1-α*, *RPB1*, and *RPB2* sequence data. The phylogeny was inferred using the IQ-tree. The maximum likelihood bootstrap valued greater than 60% (on the left) and the Bayesian posterior probabilities over 0.6 (on the right) were indicated above the nodes. The new species and known ones were indicated in back bold font.

**Figure 5 jof-10-00297-f005:**
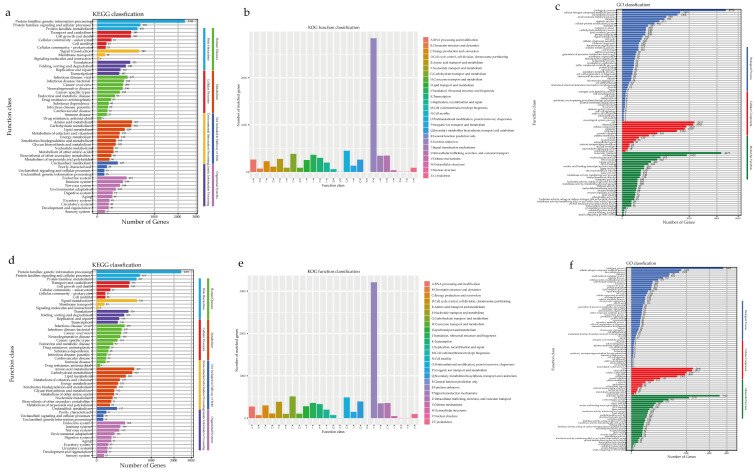
Functional annotation of *Pe. elaphomyceticola* and *Pl. fusiformispora* genes encoding the proteins. (**a**–**c**) *Pe. elaphomyceticola* (**a**) KEGG, (**b**) KOG, (**c**) GO; (**d**–**f**) *Pl. fusiformispora* (**d**) KEGG, (**e**) KOG, (**f**) GO).

**Figure 6 jof-10-00297-f006:**
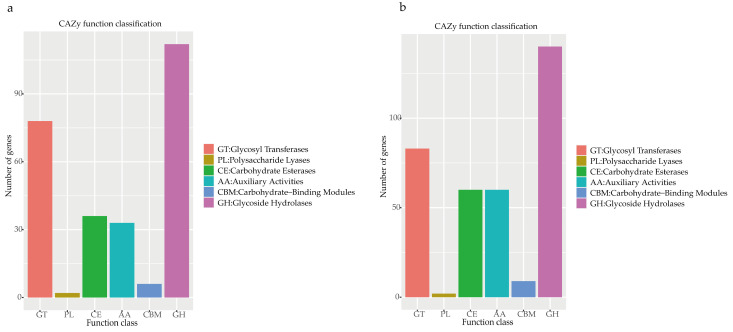
CAZy functional classification chart of *Pe. elaphomyceticola* and *Pl. fusiformispora*. (**a**) *Pe. elaphomyceticola* (**b**) *Pl. fusiformispora*.

**Figure 7 jof-10-00297-f007:**
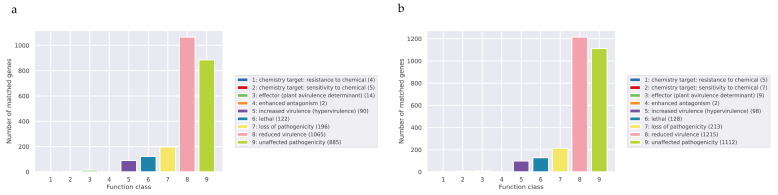
Distribution of the mutation types in the PHI phenotype of *Pe. elaphomyceticola* and *Pl. fusiformispora*. (**a**) *Pe. elaphomyceticola* (**b**) *Pl. fusiformispora*.

**Figure 8 jof-10-00297-f008:**
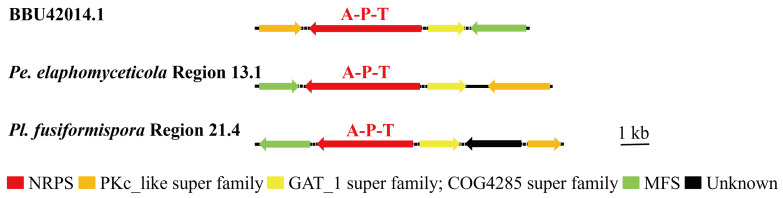
Comparison of putative BGC of ε-poly lysine in *Pe. elaphomyceticola* and *Pl. fusiformispora*. The number after the region and the number before the decimal point represent the scaffold, and the number after the decimal point represents the gene cluster.

**Figure 9 jof-10-00297-f009:**
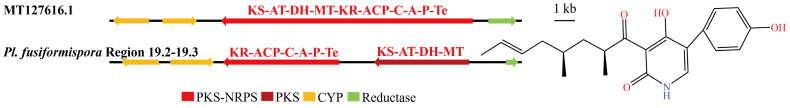
Comparison of biosynthesis of putative Tolypyridone in *Pl. fusiformispora*. The number after the region and the number before the decimal point represent the scaffold, and the number after the decimal point represents the gene cluster.

**Figure 10 jof-10-00297-f010:**
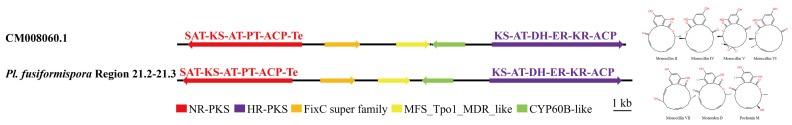
Comparison of biosynthesis of putative Monorden D/monocillin IV/monocillin VII/pochonin M/monocillin V/monocillin II in *Pl. fusiformispora*. The number after the region and the number before the decimal point represent the scaffold, and the number after the decimal point represents the gene cluster.

**Figure 11 jof-10-00297-f011:**
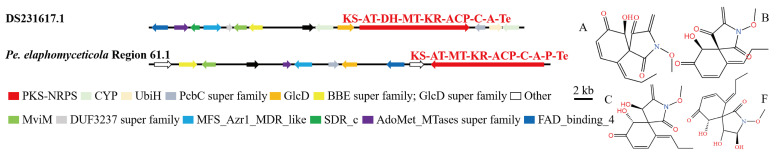
Comparison of biosynthesis of putative Triticone DABFC in *Pe. elaphomyceticola*. The number after the region and the number before the decimal point represent the scaffold, and the number after the decimal point represents the gene cluster.

**Figure 12 jof-10-00297-f012:**
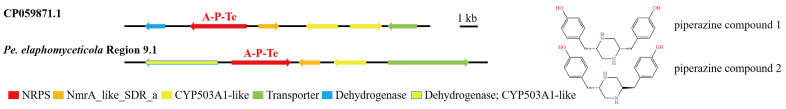
Comparison of biosynthesis of putative piperazine compound **1** and piperazine compound **2** in *Pe. elaphomyceticola*. The number after the region and the number before the decimal point represent the scaffold, and the number after the decimal point represents the gene cluster.

**Figure 13 jof-10-00297-f013:**
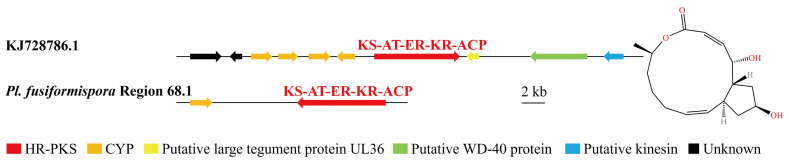
Comparison of biosynthesis of putative 4-epi-15-epi-brefeldin A in *Pl. fusiformispora*. The number after the region and the number before the decimal point represent the scaffold, and the number after the decimal point represents the gene cluster.

**Figure 14 jof-10-00297-f014:**
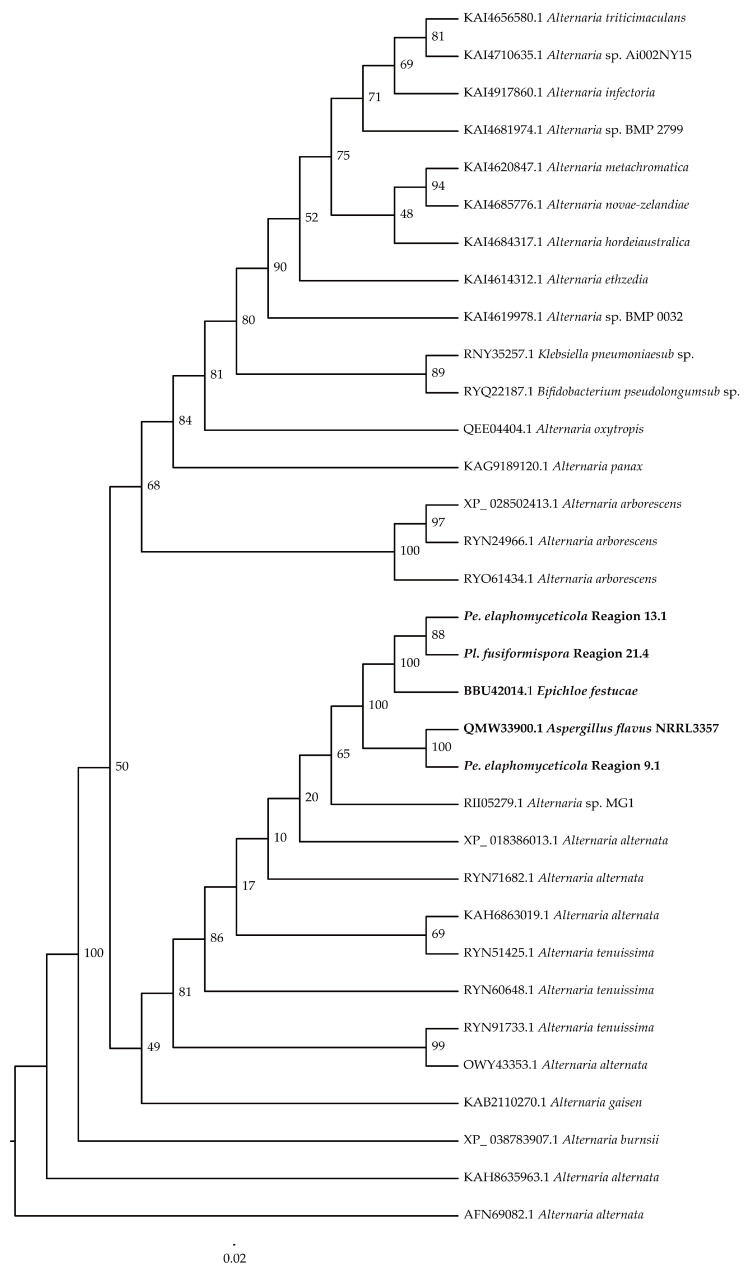
Clustering tree of NRPS proteins between *Pe. elaphomyceticola* and *Pl. fusiformispora* species and other fungi. Values at the nodes represent bootstrap values. Bold lines are shown at the nodes for 100 support. The scale bar 2.0 indicates the number of expected mutations per site. Bold indicates their clustering situation.

**Figure 15 jof-10-00297-f015:**
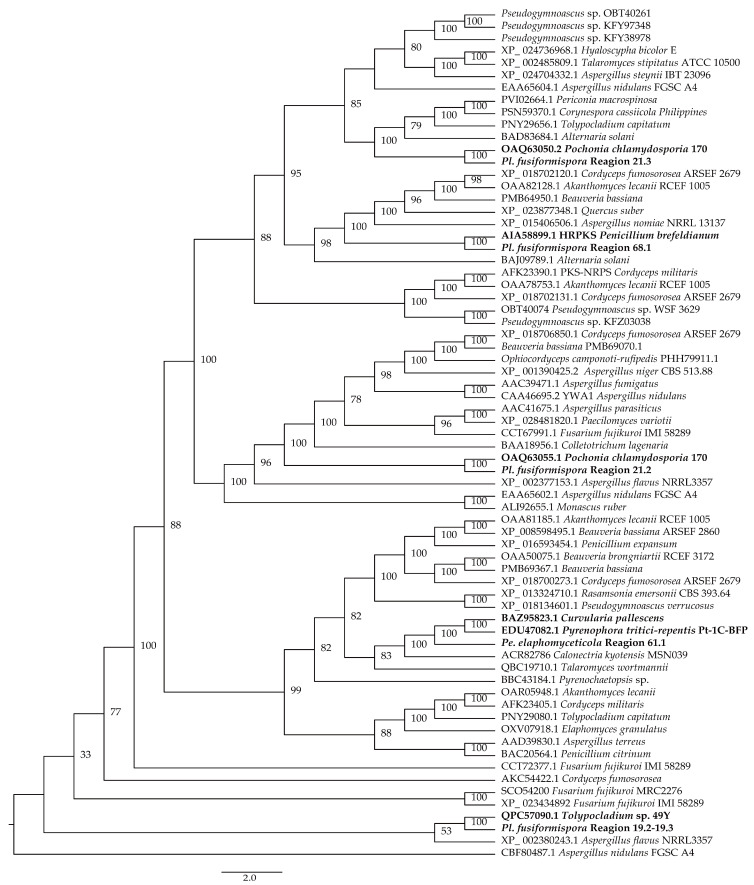
Clustering tree of hybrid PKS–NRPS proteins between *Pe. elaphomyceticola* and *Pl. fusiformispora* species and other fungi. Values at the nodes represent bootstrap values. Bold lines are shown at the nodes for 100 support. The scale bar 2.0 indicates the number of expected mutations per site. Bold indicates their clustering situation.

**Figure 16 jof-10-00297-f016:**
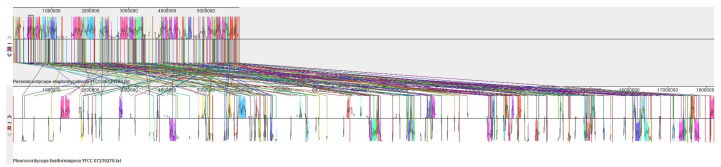
Scaffold synteny analysis of biosynthetic gene clusters containing secondary metabolites in the genomes of *Pe. elaphomyceticola* and *Pl. fusiformispora*. Numerical representation of gene length (bp). Different colored lines represent collinear regions between different genomes.

**Table 1 jof-10-00297-t001:** Sources of selected isolates and GenBank accession number for ITS and five genes of three genera within Polycephalomycetaceae in this study.

Species Name	Voucher	ITS	SSU	LSU	*TEF-1α*	*RPB1*	*RPB2*	References
*Cordyceps pleuricapitata*	NBRC 109978	AB925940		AB925977				Unpublished
*Cordyceps pleuricapitata*	NBRC 109977	AB925939		AB925976				Unpublished
*Cordyceps pleuricapitata*	NBRC 100746	JN943306	JN941749	JN941392	KF049680	JN992483	KF049668	[26]
*Pleurocordyceps parvicapitata*	MFLU 21-0270	OQ172082	OQ172105	OQ172054	OQ459722	OQ459751	OQ459796	[1]
*Pleurocordyceps sinensis*	MFLU 21-0269	OQ172080	OQ172122	OQ172050	OQ459742	OQ459768		[1]
*Pleurocordyceps sinensis*	GACP 20-2305	OQ172075	OQ172108	OQ172045	OQ459725	OQ459753	OQ459799	[1]
*Pleurocordyceps sinensis*	GACP 19-2301	OQ172078	OQ172124	OQ172053	OQ459744		OQ459816	[1]
*Pleurocordyceps sinensis*	GACP 20-2304	OQ172074	OQ172107	OQ172044	OQ459724		OQ459798	[1]
*Pleurocordyceps sinensis*	GZU 20-0865	OQ172071	OQ172096	OQ172043	OQ459713			[1]
*Pleurocordyceps sinensis*	MFLU 21-0268	OQ172070	OQ172123	OQ172052	OQ459743		OQ459815	[1]
*Pleurocordyceps fusiformispora*	YFCC 07239279	PP002030		PP410610	PP254877	PP581807	PP581824	This study
*Pleurocordyceps fusiformispora*	YFCC 07239280	PP002031		PP410611	PP254878	PP581808	PP581825	This study
*Pleurocordyceps fusiformispora*	YFCC 07319281				PP254879	PP581809	PP581826	This study
*Pleurocordyceps vitellina*	KUMCC 3006	OQ172089		OQ172061	OQ459729	OQ459757	OQ459803	[1]
*Pleurocordyceps vitellina*	KUMCC 3007	OQ172090		OQ172062	OQ459730	OQ459758	OQ459804	[1]
*Pleurocordyceps agarica*	YHHPA 1305^T^	KP276651	KP276655		KP276659	KP276663	KP276667	[25]
*Pleurocordyceps agarica*	YHCPA1307	KP276654	KP276658		KP276662	KP276666	KP276670	[25]
*Pleurocordyceps agarica*	YHCPA 1303	KP276653	KP276657		KP276661	KP276665	KP276669	[25]
*Pleurocordyceps aurantiaca*	MFLUCC 17-2113^T^	MG136916	MG136904	MG136910	MG136875	MG136866	MG136870	[27]
*Pleurocordyceps aurantiaca*	MFLUCC 17-2114	MG136917	MG136905	MG136911	MG136874		MG136871	[27]
*Pleurocordyceps aurantiaca*	MFLU 17-1393^T^		MG136907	MG136913	MG136877	MG136868	MG136873	[27]
*Pleurocordyceps formosus* like	NBRC 101760	MN586827	MN586818	MN586836	MN598051	MN598042	MN598060	[14]
*Pleurocordyceps formosus* like	NBRC 109984	MN586828	MN586819	MN586837	MN598052	MN598043		[14]
*Pleurocordyceps formosus* like	NBRC 109985	MN586829	MN586820	MN586838	MN598053	MN598044		[14]
*Pleurocordyceps heilongtanensis*	KUMCC 3008	OQ172091	OQ172111	OQ172063	OQ459731	OQ459759	OQ459805	[1]
*Pleurocordyceps kanzashianus*		AB027371	AB027325	AB027371				[28]
*Pleurocordyceps lanceolatus*	GACP 17-2004^T^	OQ172076	OQ172110	OQ172046	OQ459726	OQ459754	OQ459800	[1]
*Pleurocordyceps lanceolatus*	GACP 17-2005^T^		OQ172109	OQ172047	OQ459727	OQ459755	OQ459801	[1]
*Pleurocordyceps lianzhouensis*	HIMGD20918^T^	EU149921	KF226245	KF226246	KF226248	KF226247		[29]
*Pleurocordyceps lianzhouensis*	GIMYY9603	EU149922	KF226249	KF226250	KF226252	KF226251		[29]
*Pleurocordyceps marginaliradians*	MFLU 17-1582^T^	MG136920	MG136908	MG136914	MG136878	MG136869	MG271931	[27]
*Pleurocordyceps marginaliradians*	MFLUCC 17-2276^T^	MG136921	MG136909	MG136915	MG136879		MG271930	[27]
*Pleurocordyceps* *nipponica*	BCC 1682	KF049664	KF049620	KF049638	KF049694			[26]
*Pleurocordyceps* *nipponica*	BCC 18108	KF049657	MF416624	MF416569	MF416517	MF416676	MF416462	[26]
*Pleurocordyceps nipponica*	BCC 1881		KF049618	KF049636	KF049692		KF049674	[26]
*Pleurocordyceps nutansis*	GACP 19-1906	OQ172079	OQ172117	OQ172049	OQ459737	OQ459763	OQ459809	[1]
*Pleurocordyceps nutansis*	GACP 19-1907	OQ172087	OQ172118	OQ172059	OQ459738	OQ459764	OQ459810	[1]
*Pleurocordyceps nutansis*	GACP 19-3019^T^	OQ172086	OQ172120	OQ172058	OQ459740	OQ459766	OQ459812	[1]
*Pleurocordyceps nutansis*	MFLU 21-0275^T^	OQ172073	OQ172119	OQ172048	OQ459739	OQ459765	OQ459811	[1]
*Pleurocordyceps onorei*	BRA CR23904	KU898843						[30]
*Pleurocordyceps onorei*	BRA CR23902^T^	KU898841						[30]
*Pleurocordyceps parvicapitata*	MFLU 21-0271^T^	OQ172083	OQ172106	OQ172055	OQ459723	OQ459752	OQ459797	[27]
*Pleurocordyceps parvicapitata*	MFLU 21-0272	OQ172084	OQ172099	OQ172056	OQ459716	OQ459745	OQ459790	[1]
*Pleurocordyceps phaothaiensis*	BCC84553^T^	MF959733		MF959737	MF959742	MF959745		[30]
*Pleurocordyceps phaothaiensis*	BCC84552	MF959732		MF959736	MF959740	MF959744		[30]
*Pleurocordyceps phaothaiensis*	BCC84551	MF959731		MF959735	MF959739	MF959743		[30]
*Pleurocordyceps ramosopulvinata*	EFCC 5566			KF049627	KF049682	KF049645		[26]
*Pleurocordyceps ramosopulvinata*	SU 65			DQ118742	DQ118753	DQ127244		[31]
*Pleurocordyceps sinensis*	CN 80-2^T^	HQ832884	HQ832887	HQ832886	HQ832890	HQ832888	HQ832889	[32]
*Pleurocordyceps sinensis*		HQ918290						[33]
*Pleurocordyceps sinensis*	HMAS 43720^T^	NR_119928		NG_042573				[32]
*Pleurocordyceps* sp.	BCC 2637	KF049663		KF049637	KF049693		KF049675	[26]
*Pleurocordyceps* sp.	JB07.08. 16_08	KF049662	KF049616	KF049635	KF049690	KF049652	KF049672	[26]
*Pleurocordyceps* sp.	JB07.08. 17_07b		KF049617		KF049691	KF049653	KF049673	[26]
*Pleurocordyceps* sp.	NBRC 109987			AB925983				[14]
*Pleurocordyceps* sp.	NBRC 109988			AB925984				[14]
*Pleurocordyceps* sp.	NBRC 109990			AB925968				[14]
*Pleurocordyceps* sp.	NBRC 110224			AB925969				[14]
*Pleurocordyceps tomentosus*	BL4	KF049666	KF049623	KF049641	KF049697	KF049656	KF049678	[26]
*Pleurocordyceps vitellina*	KUMCC 3005	OQ172088		OQ172060	OQ459728	OQ459756	OQ459802	[1]
*Pleurocordyceps yunnanensis*	YHCPY1005	KF977848			KF977850	KF977852	KF977854	[24]
*Pleurocordyceps yunnanensis*	YHHPY1006^T^	KF977849			KF977851	KF977853	KF977855	[24]
*Perennicordyceps elaphomyceticola*	MFLU 21-0262	OQ172064	OQ172101	OQ172032	OQ459718	OQ459747	OQ459792	[1]
*Perennicordyceps cuboidea*	NBRC 103836	JN943332	JN941721	JN941420	AB972951	JN992455	AB972955	[34]
*Perennicordyceps cuboidea*	NBRC 103834	JN943330	JN941723	JN941418		JN992457		[34]
*Perennicordyceps cuboidea*	NBRC 103835	JN943333	JN941722	JN941419		JN992456		[34]
*Perennicordyceps cuboidea*	NBRC 101740	JN943331	JN941724	JN941417	KF049684	JN992458		[34]
*Perennicordyceps cuboidea*	CEM 1514		KF049609	KF049628	KF049683			[26]
*Perennicordyceps elaphomyceticola*	MFLU 21-0264	OQ172067	OQ172103	OQ172035	OQ459720	OQ459749	OQ459794	[1]
*Perennicordyceps elaphomyceticola*	MFLU 21-0263	OQ172065	OQ172102	OQ172033	OQ459719	OQ459748	OQ459793	[1]
*Perennicordyceps elaphomyceticola*	YFCC 06129282	PP002336		PP024253	PP035749	PP581810	PP581823	This study
*Perennicordyceps paracuboidea*	NBRC 100942	JN943337	JN941711	JN941430		JN992445	AB972958	[34]
*Perennicordyceps prolifica*	TNS-F-18547	KF049660	KF049613	KF049632	KF049687	KF049649	KF049670	[26]
*Perennicordyceps prolifica*	TNS-F-18481	KF049659	KF049612	KF049631	KF049686	KF049648		[26]
*Perennicordyceps ryogamiensis*	NBRC 101751	JN943343	JN941703	JN941438	KF049688	JN992437		[34]
*Perennicordyceps ryogamiensis*	NBRC 103837	JN943346	JN941702	JN941439		JN992436		[34]
*Perennicordyceps ryogamiensis*	NBRC 103842	JN943345	JN941701	JN941440		JN992435		[34]
*Polycephalomyces formosus*	GACP 21-WFKQ03	OQ172094	OQ172113	OQ172039	OQ459733			[1]
*Polycephalomyces formosus*	GACP 21-WFKQ04	OQ172095	OQ172114	OQ172040	OQ459734			[1]
*Polycephalomyces albiramus*	GACP 21-XS08^T^	OQ172092	OQ172115	OQ172037	OQ459735	OQ459761	OQ459807	[1]
*Polycephalomyces albiramus*	GACPCC 21-XS08^T^	OQ172093	OQ172116	OQ172038	OQ459736	OQ459762	OQ459808	[1]
*Polycephalomyces* *formosus*	NBRC 100686	MN586830	MN586821	MN586839	MN598054	MN598045	MN598061	[14]
*Polycephalomyces* *formosus*	NBRC 100687	MN586831	MN586822	MN586840	MN598055	MN598046	MN598062	[14]
*Polycephalomyces* *formosus*	NBRC 109994	MN586834	MN586825	MN586843	MN598058	MN598049	MN598065	[14]
*Tolypocladium ophioglossoides*	NBRC 100998	JN943319	JN941735	JN941406	AB968602	JN992469	AB968563	[35]
*Tolypocladium ophioglossoides*	NBRC 106330	JN943321	JN941734	JN941407	AB968603	JN992468	AB968564	[35]

## Data Availability

Data are contained within the article and Appendix A.

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
