# Peer review of "Molecular Phylogenetic and Comparative Genomic Analysis of Pleurocordyceps fusiformispora sp. nov. and Perennicordyceps elaphomyceticola in the Family Polycephalomycetaceae"

_jof, 2024, doi:10.3390/jof10040297_

Round 1
Reviewer 1 Report
Generally the manuscript "Molecular Phylogenetic and Comparative Genomic Analysis of Perennicordyceps elaphomyceticola and Pleurocordyceps fusiformispora in the Family Polycephalomycetaceae" is well written and suitable for publication in JoF.
See section Materials and Methods, as Results section of this review.
Author Response
Responses to the reviewer comments
Dear reviewer
Thank you very much for your insightful and helpful comments on our manuscript. We
have revised the paper according to your comments. Our response to your suggestion is
as follows: (marked with color yellow in the manuscript)
Responds to the reviewer’s comment:
Generally the manuscript "Molecular Phylogenetic and Comparative Genomic Analysis
of Perennicordyceps elaphomyceticola and Pleurocordyceps fusiformispora in the Family
Polycephalomycetaceae" is well written and suitable for publication in JoF.
Response : We thank the reviewers for their suggestions and patient comments. These
comments are all valuable and very helpful for revising and improving our paper, as well
as the important guiding significance to our research. The following are our responses to
all questions one by one.
Comment 1: The authors need to make sure that they provide all of the relevant
formation in the materials and method section. For example, PDA media (Potato
Dextrose Agar, PDA (Supplier). The authors also need to provide the correct supplier
information. For example, ZR Fungal DNA Kit (Catalogue number D6005). The authors
need to clearly explain how the genome sequencing was performed. As it is written now
it is difficult to follow. How exactly did the authors filter and trimmed the reads? Also
please provide references for the software used. The software fastp (version 0.21.0) was
used to filter the raw reads, discard low-quality reads, and obtain clean data. In order to
obtain high-quality collication-free genomic assembly readings, AdapterRemoval
(version 2) and SOAPec (version 2) software were used to filter the raw data. Please
provide references for the sequences used, as well as the names of the genes. Also,
reference and name NCBI correctly. Clustal W program in the MEGA5.0 software? "The
polygene nucleotide sequences (ITS, SSU, LSU, TEF-1a, RPB1, RPB2) were downloadedfrom NCBI and compared with these sequences using the Clustal W program in the
MEGA5.0 software for multi-sequence comparison."
Response 1: Thanks for the valuable review. We have made modifications. PDA solid
media (20 g/L potato powder, 20 g/L glucose, 18 g/L agar powder, 1 L H2O (All chemicals
& reagents were from Yunnan, China)). (Please see the Revised Manuscript, line
86,87,94,95).
Thank you very much for your advice! total genomic DNA was extracted using the Plant
DNA Isolation Kit (Foregene Co., Ltd., Chengdu, China).We have provided the correct supplier
information.(Please see the Revised Manuscript, line 128,129).
Thank you very much for your suggestion! We have improved the description of
genome sequencing methods. Illumina NovaSeq 2000 (Nanopore, Wuhan, China)
high-throughput sequencing platform used for sequencing a gene library with 400 bp
insertion
fragment,
Sequencing
Mode:
Paired-end,
2x150bp.
The
fastp
([https://github.com/OpenGene/fastp]) was used to filter the raw reads, discard
low-quality reads, and obtain clean data. In order to ensure the quality of subsequent
information analysis, it is necessary to further filter the raw data to generate high-quality
sequences. The standards for data filtering mainly include the following points: Joint
contamination removal, using AdapterRemoval (v2.0) [60] to remove joint contamination
from the 3 'end; Quality correction, using SOAPec (v2.0) software to perform quality
correction on all reads based on Kmer frequency, with a Kmer setting of 17 used for
correction. Some software reference materials were not found. We have made
modifications in the manuscript and highlighted them in yellow. (Please see the Revised
Manuscript, line 132-142).
Thanks for this suggestion! These methods: Clustal W program in the MEGA5.0
software." The polygene nucleotide sequences (ITS, SSU, LSU, TEF-1a, RPB1, RPB2) were
downloaded from NCBI and compared with these sequences using the Clustal W
program in the MEGA5.0 software for multi-sequence comparison. We referred to the
methods of [29,30]. (Please see the Revised Manuscript, line 163-166).Comment 2: There are missing data in Table 1. Why is this? The authors need to submit
their sequences to Genbank and should have information for data generated in this study.
Generally the figures are good. Especially the photos. However, it is impossible to read
Figures 5-7. The authors can increase the size of of these figures? Figures 8 to 11 is
distorted and difficult to read.
Response 1: Thanks for the review. The data has been submitted to the NCBI. (Please see
the Revised Manuscript, Table 1). Unclear images have been replaced. (Please see the
Revised Manuscript, Figures 5-7; Figures 8-11).
Figure 5. Functional annotation of Pe. elaphomyceticola and Pl. fusiformispora genes
encoding the proteins. a,b,c Pe. elaphomyceticola (a: KEGG b: KOG c: GO); d,e,f Pl.
fusiformispoa (d: KEGG e: KOG f: GO).Figure 6. CAZy functional classification chart of Pe. elaphomyceticola and Pl. fusiformispora.
(a Pe. elaphomyceticola b Pl. fusiformispora).
Figure 7. Distribution of the mutation types in the pathogen-PHI phenotype of Pe.
elaphomyceticola and Pl. fusiformispora. (a Pe. elaphomyceticola b Pl. fusiformispora).
Figure 8. Comparison of putative BGC of ε-poly lysine in Pe. elaphomyceticola and Pl.
fusiformispora. The number after the region and the number before the decimal point
represent the scaffold, and the number after the decimal point represents the gene
cluster.Figure 9. Comparison of biosynthesis of putative Tolypyridone in Pl. fusiformispora. The
number after the region and the number before the decimal point represent the scaffold,
and the number after the decimal point represents the gene cluster.
Figure 10. Comparison of biosynthesis of putative Monorden D/monocillin IV/monocillin
VII/pochonin Mmonocillin V/monocillin II in Pl. fusiformispora. The number after the
region and the number before the decimal point represent the scaffold, and the number
after the decimal point represents the gene cluster.
Figure 11. Comparison of biosynthesis of putative Triticone DABFC in Pe.
elaphomyceticola. The number after the region and the number before the decimal point
represent the scaffold, and the number after the decimal point represents the gene
cluster.
Best regard!
Hong Yu

Reviewer 2 Report
This manuscript contains very valuable research results obtained as a result of numerous well-conducted scientific experiments. Based on morphology and molecular phylogeny, a new species Pleurocordyceps fusiformispora was described and a new taxonomic combination for Perennicordyceps elaphomyceticola was established. There are many research results on these two species, which included genome sequencing, gene prediction and secondary metabolites. For these reasons, these results should be published in JoF/MDPI. Unfortunately, there are numerous errors in the manuscript. Most of these errors are minor and easy to correct. However, in some places the text is imprecise and therefore difficult to understand, and in some places the sentence structure is incorrect. The taxonomic part is well prepared but requires some additions and clarifications. There are also numerous mistakes in the names of fungi. Since the numbered lines are only from the middle of the manuscript, the remarks are written on the attached PDF
see attached pdf

Author Response
Responses to the reviewer comments
Dear reviewer
Sincerely thanks for your valuable suggestions. We realized that we should have a more rigorous attitude towards the manuscript. The manuscript has been revised according to your comments, and the answers are as follows: (marked with color green in the manuscript)
Responds to the reviewer’s comment:
This manuscript contains very valuable research results obtained as a result of numerous well-conducted scientific experiments. Based on morphology and molecular phylogeny, a new species Pleurocordyceps fusiformispora was described and a new taxonomic combination for Perennicordyceps elaphomyceticola was established. There are many research results on these two species, which included genome sequencing, gene prediction and secondary metabolites. For these reasons, these results should be published in JoF/MDPI. Unfortunately, there are numerous errors in the manuscript. Most of these errors are minor and easy to correct. However, in some places the text is imprecise and therefore difficult to understand, and in some places the sentence structure is incorrect. The taxonomic part is well prepared but requires some additions and clarifications. There are also numerous mistakes in the names of fungi. Since the numbered lines are only from the middle of the manuscript, the remarks are written on the attached PDF
Response : We thank the reviewers for their suggestions and patient comments. We greatly appreciate the reviewer's contribution to language editing. The following are our responses to all questions one by one.
Comment 1: Article: genus name is only partially written in Italic. add sp. nov. and replace before Perennicordyceps elaphomyceticola (as it is presented in the taxonomic section)
Response 1: Many thanks for your suggestion! Has been changed to: Molecular Phylogenetic and Comparative Genomic Analysis of Pleurocordyceps fusiformispora sp.nov. and Perennicordyceps elaphomyceticola in the Family Polycephalomycetaceae. We have made modifications in the manuscript and highlighted them in green. (Please see the Revised Manuscript, line 3,4 )
Comment 2: Introduction: remove the dot; add space, remove the dot;
Response 2: Thanks for the review. We have made modifications in the manuscript and highlighted them in green. (Please see the Revised Manuscript, line 35 ,37,38,57,58)
Comment 3: Introduction:incorrect name of the fungus it should be Nephromopsis pallescens it should be in italic.
Response 3: Thanks for the review. We have revised it to: Nephrmopsis pallescens. We have made modifications in the manuscript and highlighted them in green. (Please see the Revised Manuscript, line 54)
Comment 4: Introduction: the last paragraph needs to be changed. Such text should be in Abstract. Here you should state the purpose of this work.
Response 4: Thank you very much for your suggestion! We have revised it to: In this study, the species Pl. fusiformispora Hong Yu bis & Z.H. Liu, D.X. Tang, Y.L. Lu, sp. nov. was first introduced. In order to discover more potential gene clusters of SMs, whole genomes of Pl. fusiformispora and Pe. elaphomyceticola were sequenced and annotated, as well as used in gene mining studies. The potential of the Polycephalomycetaceae fungi to produce the SMs was further analyzed. (Please see the Revised Manuscript, line 64,65,66,67,68)
Comment 5: Materials and Methods: add space.
Response 5: Thanks for the review. We have made modifications in the manuscript and highlighted them in green. (Please see the Revised Manuscript, line 103)
Comment 6: Materials and Methods: replace it with numbers. (Vilgalys and Hester, 1990; Hopple, 1994), (Rehner and Buckley 2005)
Response 6: Thanks for this suggestion! We have made modifications in the manuscript and highlighted them in green. (Please see the Revised Manuscript, line 105,106)
Comment 7: Phylogenetic Analysis: This was consistent with the results [1]. this belongs to discussion
Response 7: Thanks for the review. However, We believe that this is an analysis of the results of tree construction, and does not require further discussion.​
Comment 8: Table 1: it should be Pleurocordyceps; italic; sp. it should be not italic, also in other places;
Response 8: Thanks for the review. We have made modifications in the manuscript and highlighted them in green. (Please see the Revised Manuscript Table 1)
Comment 9: Taxonomy: it should be in italic and Pleurocordyceps fusiformispora.
Response 9: Thank you very much for your suggestion. We have revised it to: Pleurocordyceps fusiformispora. We have made modifications in the manuscript and highlighted them in green. (Please see the Revised Manuscript, line 206).
Comment 10: Taxonomy: not clear: Hong Yu bis
Response 10: Thanks for the review. In the early days, there was an author who published new plant species, also named Hong Yu, and there was another author who published new fungi species, named Hui Yu, abbreviated as H. Yu. In order to distinguish the two authors, the author Hong Yu of this manuscript, according to the International Code of Nomenclature, the author of this manuscript Hong Yu is followed by bis.
Comment 11: Taxonomy: add space such errors should be corrected throughout the manuscript; it should be in italic such errors should be corrected throughout the manuscript; Sexualmorph ?? Simple: not clear;
Response 11: Many thanks for your suggestion! We have made modifications in the manuscript and highlighted them in green. (Please see the Revised Manuscript, line 210,211,213).
Comment 12: Taxonomy: point out the differences between both types(α- and β-phialides)
Response 12: Thanks for the review. The α-phialides tapering abruptly from the base to the apex,narrow lageniform or subulate. The β-phialides tapering gradually from the base to the apex,lanceolate. We have made modifications in the manuscript and highlighted them in green. (Please see the Revised Manuscript, line 220,221,222,223).
Comment 13: Figure 2a - requires detailed explanations of each of the visible morphological elements.
Response 13: Many thanks for your valuable advice! We have made modifications in the manuscript and highlighted them in green. (Please see the Revised Manuscript Figure 2a).
Figure 2. Pleurocordyceps fusiformispora (Holotype: YFCC 07239279) a Overview of Pleurocordyceps fusiformispora and its host. b, c Colony obverse and reverse. d, e, f, g, h α-phialides and α-conidia. i, j, k β-phialides and β-conidia. Scale Bars: a=1cm; b, c=2 cm; d, e, f, h, i, j=10 um; g=20 um; k=5 um.
Comment 14: d: α-conidia, e: α-philakdes; whether phialides and conidia are not visible on both
Response 14: Thanks for the review. We have made modifications in the manuscript and highlighted them in green. (Please see the Revised Manuscript, line 241,242).
Comment 15: Table 2 should be moved to Supplementary material. Table 2: numerous faults need to be corrected. Table 2: cm??-at the front of the table you write that the dimensions are in mm, why here are cm. Table 2: it should be sp.
Response 15: Thanks for this suggestion! We have made modifications in the manuscript and highlighted them in green. (Please see the Revised Manuscript Supplementary material S4).
Comment 16: Perennicordyceps elaphomyceticola in italic! comb. Nov?
Response 16: Thanks for the review. We have revised it to: Perennicordyceps elaphomyceticola; comb.nov. We have made modifications in the manuscript and highlighted them in green. (Please see the Revised Manuscript, line 243, 244).
Comment 17: ??according to what rules the text is written;according to what rules the text is written?
Response 17: Thank you very much for your valuable advice! We have rewritten this section. Etymology: The specific epithet elaphomyceticola is based on the host genus, from which the fungus was isolated. Parasitic on Elaphomyces sp. (Elaphomycetaceae), from soil. Sexual morph: Stromata 5.1–6.2 cm long, 0.5–0.7 cm wide, cylindrical, solitary or several, branched, the color gradually becoming lighter towards the apex, yellow to dark yellow to light yellow, hard. Fertile heads 1.5-2 cm long, 0.1–0.3 cm wide, branched, dark yellow to light yellow, upper surface roughened. Perithecia 259–519 × 152–291 μm, superficial, ovoid to ellipsoid. Asci 164–173 × 3.1–5.5 μm, hyaline, cylindrical. Apical cap 2.1–3.5×3.6–4.2μm, thin, hyaline.Ascospores 55.1–105× 0.8–1.2 μm, irregular multiseptate. Secondary spores 0.8–1.1× 0.6–0.8 μm globose to cylindrical, onecelled, hyaline, smooth-walled. Asexual morph: (see Figure 4) Colonies on PDA 3.7–4.0 cm in diameter after 40 days at 25 ℃,usually verrucose, white to orange-yellow. Reverse appearing vague concentric rings, black-brown in the centre, maple-colored at the edge. Phialides developing from the edge of the colony and conidial mass of the synnema; phialides cylindrical to subulate at the base, occurring directly on the aerial hyphae, 16.8–31.9 µm in length, tapering gradually from 2.0–3.8 µm at the base to 0.4–1.1 µm at the apex, generating the single or lumpy conidia. Conidia oval 3.2–5.1 × 0.4–1.2 µm. We have made modifications in the manuscript and highlighted them in green. (Please see the Revised Manuscript, line 247-262).
Comment 18: Why is there no mention of stromata in the description, but stromata are presented in Figure 3? Additional information must be made. The figure should match the description.
Response 18: Thanks for this suggestion! We have revised it to: Stromata 5.1–6.2 cm long, 0.5–0.7 cm wide, cylindrical, solitary or several, branched, the color gradually becoming lighter towards the apex, yellow to dark yellow to light yellow, hard. Fertile heads 1.5–2 cm long, 0.1–0.3 cm wide, branched, dark yellow to light yellow, upper surface roughened. This was due to our mistake. We have made modifications in the manuscript and highlighted them in green. (Please see the Revised Manuscript, line 249,250,251,252).
Comment 19: fliform??
Response 19: Thanks for the review. We have revised it to: cylindrical. We have made modifications in the manuscript and highlighted them in green. (Please see the Revised Manuscript, line 253).
Comment 20: Phialides of only one type not clear ??
Response 20: Thanks for this suggestion! We have revised it to: Phialides developing from the edge of the colony and conidial mass of the synnema; phialides cylindrical to subulate at the base, occurring directly on the aerial hyphae, 16.8–31.9 µm in length, tapering gradually from 2.0–3.8 µm at the base to 0.4–1.1 µm at the apex, generating the single or lumpy conidia. We have made modifications in the manuscript and highlighted them in green. (Please see the Revised Manuscript, line 259–262).
Comment 21: above it says Elaphomyces muricatus-which information is correct?
Response 21: Thanks for this suggestion! We have revised it to: Elaphomyces sp. We have made modifications in the manuscript and highlighted them in green. (Please see the Revised Manuscript, line 264).
Comment 22: Each element in Figure 3a-3d should be described in detail with drawn arrows.
Response 22: Thanks for the review. We have provided a detailed description using the plotted straight lines. (Please see the Revised Manuscript Figure 3a-3d).
Figure 3. Sexual morph of the Perennicordyceps elaphomyceticola (YFCC 06129282). a Stromata emerging from infected Elaphomyces sp. b Fertile head of ascostroma. c Vertical section of stroma. d Perithecia. e-h: Asci. i Asci and ascospore. j, k Ascospore. l Secondary ascospores. Scale Bars: a=2 cm; b=5000 um; c=500 um; d=200 um; e, f, g, h=50 um; i= j=k=20 um; l=5 um.
Comment 23: you write secondary spores in the text-here secondary ascospores. You should be aware that this is a fundamental difference. In many other fungi, such spores are called ascoconidia.
Response 23: Thanks for the review. We believe that secondary ascospores are produced by the fracture of ascospores. We have made modifications in the manuscript and highlighted them in green. (Please see the Revised Manuscript, line 269).
Comment 24: the ascospores in figure 3l are out of focus.
Response 24: Thanks for the review. We have replaced the photograph. (Please see the Revised Manuscript Figure 3).
Comment 25: Figure 5: unfortunately, the text is unreadable
Response 25: Thanks for this suggestion! We have increased the text resolution. (Please see the Revised Manuscript Figure 5).
Figure 5. Functional annotation of Pe. elaphomyceticola and Pl. fusiformispora genes encoding the proteins. a,b,c Pe. elaphomyceticola (a: KEGG b: KOG c: GO); d,e,f Pl. fusiformispora (d: KEGG e: KOG f: GO).
Comment 26: Figure 5: It should be Pl. fusiformispora
Response 26: Thanks for the review. We have revised it to: Pl. Fusiformispora. We have made modifications in the manuscript and highlighted them in green. (Please see the Revised Manuscript, line 317,318).
Comment 27: Figure 6 6a? Figure 6 6b: rather 6b?
Response 27: Thanks for this suggestion! We have made modifications in the manuscript and highlighted them in green. (Please see the Revised Manuscript, line 325,326).
Comment 28: Pathogen Host Interactions?
Response 28: Thanks for the review. The full name of PHI is Pathogen Host Interactions Database, which is a database of pathogen host interactions mainly derived from fungi, oomycetes, and bacterial pathogens. We have made modifications in the manuscript and highlighted them in green. (Please see the Revised Manuscript, line 333-335).
Comment 29: The text requires correction. What is meant by the term pathogen here? Some introduction is needed as to what this is about.
Response 29: Thanks for this suggestion! The infected hosts include animals, plants, fungi, and insects. We have made modifications in the manuscript and highlighted them in green. (Please see the Revised Manuscript, line 336-337).
Comment 30: Figure 7 7a ? remove 7
Response 30: Thanks for this suggestion! We have made modifications in the manuscript and highlighted them in green. (Please see the Revised Manuscript, line 339).
Comment 31: unaffected pathogenicity and the loss of pathogenicity gene-this requires explanation this is a contradiction.
Response 31: Thanks for the review. We have revised it to: The results showed that Pe. elaphomyceticola (Figure 7a) and Pl. fusiformispora (Figure 7b) had more reduced virulence and the loss of pathogenicity gene. We have made modifications in the manuscript and highlighted them in green. (Please see the Revised Manuscript, line 339).
Comment 32: Pathogenicity and virulence should be precisely defined in methods or in Discussion (there are different definitions in the literature). This is especially important for fungi that infect insects.
Response 32: Thanks for this suggestion! The full name of PHI is Pathogen Host Interactions, which includes experimentally validated virulence, pathogenicity, and effector genes from fungal, oomycete, and bacterial pathogens that infect animal, plant, fungal, and insect hosts. Therefore, we believe that our current data cannot accurately define pathogenicity and toxicity in discussions and results.
Comment 33: why are these fungi called in other places 'hyperparasites' and 'pathogens' here?
Response 33: Thanks for this suggestion! Because of infected hosts include fungi, and insects.
Comment 34: Pe. elaphomyceticola and Pl. Fusiformispora??
Response 34: Many thanks for your suggestion! We have made modifications in the manuscript and highlighted them in green. (Please see the Revised Manuscript, line 347).
Comment 35: Both ?? And ?
Response 35: Thanks for the review. We have made modifications in the manuscript and highlighted them in green. (Please see the Revised Manuscript, line 361,363).
Comment 36: the text with the use of literature should be moved to Discussion.
Response 36: Thanks for this suggestion! We believe it is clearer to cite the references here.
Comment 37: add space in many places in this manuscript.
Response 37: Many thanks for your suggestion! We have added spaces in many places in this manuscript. We have made modifications in the manuscript and highlighted them in green.
Comment 38: Figure 14: mistakes in the names of fungi should be corrected.
Response 38: Thanks for the review. We have corrected the error in the fungal name. (Please see the Revised Manuscript Figure 14).
Comment 39: Figure 15: mistakes in the names of fungi should be corrected.
Response 39: Thanks for the review. We have corrected the error in the fungal name. (Please see the Revised Manuscript Figure 15).
Comment 40: the text in this paragraph requires thorough tidying up.
Response 40: Many thanks for your valuable advice! We have rewritten this section. Herein, we describe a new species of Pl. fusiformispora using a combination of morphology and phylogeny. The newly established species distinctly form independent clades in the phylogenetic tree (Figure 1). Morphologically, the new species Pl. fusiformispora is similar to Pl. aurantiacus, Pl. agarica, Pl. heilongtanensis, Pl. lanceolatus, Pl. marginaliradians, Pl. nutansis, Pl. vitellina, and Pl. yunnanensis in that they have two types of phialides and conidia. However, the specials Pl. fusiformispora colonies flaky, and α-conidia ovoid or elliptic, β-conidia fusiform or long fusiform differ from those of other species of Pleurocordyceps [1, 29, 30, 44]. The discovery of the new species of Pl. fusiformispora adds to the diversity of the genus Pleurocordyceps. (Please see the Revised Manuscript, line 472-480).
Comment 41: The text requires clarification.
Response 41: Thanks for this suggestion! This refers to [26,27,28,29,30] the research results from indicate that the Polycephalomycetaceae fungi can be hyperparasitic on Cordyceps, Elaphomys, Hirsutella, Myxomycotes, and Ophiocordyceps.
Comment 42: The text needs to be changed.
Response 42: Thanks for the review. We have rewritten this section. Through this investigation, we have observed that Pl. fusiformispora can parasitize both Lepidoptera larvae and Ophiocordyceps sp. We speculate that Pl. fusiformispora may also exhibit hyperparasitism phenomenon. (Please see the Revised Manuscript, line 491-493).
Comment 43: Elaphomyces ???
Response 43: Thanks for this suggestion! We have made modifications in the manuscript and highlighted them in green. (Please see the Revised Manuscript, line 494).
Comment 44: on which host (I pay attention when the term hyperparasite is used - when the host is a parasite - does this apply to Elaphomyces?)
Response 44: Thanks for the review. We have made modifications in the manuscript and highlighted them in green. (Please see the Revised Manuscript, line 495).
Comment 45: the text requires correction
Response 45: Thanks for the review. We have made modifications in the manuscript and highlighted them in green. (Please see the Revised Manuscript, line 506-509).
Comment 46: the text requires correction
Response 46: Thanks for this suggestion! We have made modifications in the manuscript and highlighted them in green. (Please see the Revised Manuscript, line 516,517).
Comment 47: Pe. Aphomyceticola??
Response 47: Thanks for the review. We have made modifications in the manuscript and highlighted them in green. (Please see the Revised Manuscript, line 521).
Comment 48: epls ??
Response 48: Thanks for this suggestion! We have made modifications in the manuscript and highlighted them in green. (Please see the Revised Manuscript, line 526).
Comment 49: what does E. mean ???
Response 49: Thanks for the review. The meaning is endophytic fungus Epichloë festucae. We have made modifications in the manuscript and highlighted them in green. (Please see the Revised Manuscript, line 527).
Comment 50: And it was also found that Tolypyridone had antifungal activity?
Response 50: Thanks for this suggestion! We have rewritten this section. Heterologous expression in Aspergillus oryzae NSAR1 revealed that this gene cluster is functional and able to produce a rare type of 4-hydroxy pyridones called Tolypyridone. We have made modifications in the manuscript and highlighted them in green. (Please see the Revised Manuscript, line 538-540).
Comment 51: consider revisnig this sentence: And it was also found that Tolypyridone had antifungal activity.
Response 51: Thanks for this suggestion! We deleted it from the original manuscript.
Comment 52: antiviral - this information should be checked from the source.
Response 52: Thanks for the review. We have rewritten this section. Brefeldin A is a unique fungal metabolite of a 13-membered macrocyclic lactone ring [41] and shows a wide range of interesting biological activities including an inhibitory effect on virus multiplication [58,59]. We have made modifications in the manuscript and highlighted them in green. (Please see the Revised Manuscript, line 579-581).
Comment 53: AO ???
Response 53: Thanks for this suggestion! We have made modifications in the manuscript and highlighted them in green. (Please see the Revised Manuscript, line 581).
Comment 54: ??? why so many errors - did any of the authors read the text before sending it to MDPI?
Response 54: Thanks for the review. This was due to our mistake. We appreciate reviewers so much for excellent comments on our manuscript. Once again, we thank the reviewers for their hard work.
Comment 55: References - Latin names of plants, fungi and bacteria should be in Italian
Response 55: Thanks for the review. We have made modifications in the manuscript and highlighted them in green. (Please see the Revised Manuscript References).
Comment 56: error - it should be DODGE
Response 56: Thanks for this suggestion! We have made modifications in the manuscript and highlighted them in green. (Please see the Revised Manuscript, line 704).
Best regard!
Hong Yu
